# Vesicular Drug Delivery Systems: Promising Approaches in Ocular Drug Delivery

**DOI:** 10.3390/ph17040511

**Published:** 2024-04-16

**Authors:** Eslim Batur, Samet Özdemir, Meltem Ezgi Durgun, Yıldız Özsoy

**Affiliations:** 1Health Science Institute, Istanbul University, 34126 Istanbul, Türkiye; eslim.batur@iuc.edu.tr; 2Department of Pharmaceutical Technology, Faculty of Pharmacy, Istanbul University-Cerrahpaşa, 34500 Istanbul, Türkiye; 3Department of Pharmaceutical Technology, Faculty of Pharmacy, Istanbul Health and Technology University, 34445 Istanbul, Türkiye; meltem.durgun@istun.edu.tr; 4Department of Pharmaceutical Technology, Faculty of Pharmacy, Istanbul University, 34116 Istanbul, Türkiye; yozsoy@istanbul.edu.tr

**Keywords:** ocular drug delivery, vesicular systems, liposomes, targeted delivery, sustained release

## Abstract

Ocular drug delivery poses unique challenges due to the complex anatomical and physiological barriers of the eye. Conventional dosage forms often fail to achieve optimal therapeutic outcomes due to poor bioavailability, short retention time, and off-target effects. In recent years, vesicular drug delivery systems have emerged as promising solutions to address these challenges. Vesicular systems, such as liposome, niosome, ethosome, transfersome, and others (bilosome, transethosome, cubosome, proniosome, chitosome, terpesome, phytosome, discome, and spanlastics), offer several advantages for ocular drug delivery. These include improved drug bioavailability, prolonged retention time on the ocular surface, reduced systemic side effects, and protection of drugs from enzymatic degradation and dilution by tears. Moreover, vesicular formulations can be engineered for targeted delivery to specific ocular tissues or cells, enhancing therapeutic efficacy while minimizing off-target effects. They also enable the encapsulation of a wide range of drug molecules, including hydrophilic, hydrophobic, and macromolecular drugs, and the possibility of combination therapy by facilitating the co-delivery of multiple drugs. This review examines vesicular drug delivery systems, their advantages over conventional drug delivery systems, production techniques, and their applications in management of ocular diseases.

## 1. Introduction

Different kinds of ocular diseases, such as cataract, glaucoma, myopia, diabetic retinopathy (DR), age-related macular degeneration (AMD), post-operative endophthalmitis, vitreous hemorrhage, chronic dry eye, allergies, inflammations, and bacterial/fungal keratitis, affect many people around the world [1]. Eye drops, aqueous suspensions, and oily suspensions are dosage forms that are used for topical ocular drug delivery by penetrating through the cornea and distributing to the ocular tissues [1]. Even though these formulations are used to treat different types of ocular diseases, ocular lacrimation, and systemic absorption lead to poor distribution, resulting in the loss of over 95% of the dose [1]. To overcome these problems different types of administration techniques, including intravitreal (IVT) injection, subconjunctival (SCT) injection or suprachoroidal injections, and systemic delivery are used [1]. Although these methods are frequently used, they have side effects such as cataract, infection, endophthalmitis, and retinal damage [1]. Also, a tight ocular–blood barrier, an inflammatory response, and immune system activation may pose additional challenges to ocular drug delivery systems employed to treat posterior segment diseases [1,2]. All these disadvantages require alternative options for ocular drug delivery [1,2].

Vesicular drug delivery systems provide improved site specificity, stability, prolonged release, and enhanced bioavailability and penetration in ocular drug delivery [2]. Vesicular drug delivery systems like liposome, niosome, ethosome, transfersome, and others (bilosome, transethosome, cubosome, proniosome, chitosome, terpesome, phytosome, discome, and spanlastics) can encapsulate both hydrophilic and lipophilic drugs [3,4,5,6]. They also provide controlled ocular delivery by preventing the drug from being metabolized by enzymes at the tear–corneal epithelial interface and promoting a sustained and controlled release on the corneal surface [4]. 

This review aims to inform the reader and demonstrate the advantages of using vesicular systems in ocular drug delivery. There are articles in the literature on the ocular use of vesicular systems [3,6,7]. Most of these articles have been studied with liposomes [8,9]. However, in our study, different kinds of vesicular systems (liposome, niosome, ethosome, transfersome, and others (bilosome, transethosome, cubosome, proniosome, chitosome, terpesome, phytosome, discome, spanlastics, flexosomes, phytocubosome, and oleophytocubosome)) were comprehensively evaluated, including studies with different therapeutic agents, their therapeutic effects, production methods, and formulation content (lipids/surfactants). 

We will first discuss the anatomical structure of the eye and the difficulties of ocular administration, then ophthalmic applications and the use of vesicular systems in the treatment of various diseases, their advantages and disadvantages, and the structural features of vesicular systems.

## 2. Ocular Anatomy and Drug Administration

The eyes are specialized organs consisting of intricate tissue segments [2]. The ocular structure could be categorized into anterior and posterior segments [2]. These segments arise through the anatomical separation of the eyeball by the ciliary body and lens [1]. The anterior segment is principally constituted by structural elements such as tear film, cornea, pupil, conjunctiva, lens, and ciliary body, and is filled with aqueous humor [1,2]. Conversely, the posterior segment of the eye is characterized by essential constituents such as the sclera, choroid, retina, and optic nerve, and is filled with vitreous humor [1,2].

The iris (colored part of the eye) regulates the amount of light that can pass through the eye and the pupil is the black opening in the center of the iris that changes in size depending on the amount of light present [1] The ciliary body comprises ciliary muscles, stroma, ciliary epithelia, and blood capillaries [1]. It establishes communication between the anterior and posterior chambers of the eye [10]. The cornea is an optically clear portion of the eye, composed primarily of five components: epithelium, Bowman’s layer, stroma, Descemet’s membrane, and endothelium [1,10]. The eye includes a lens positioned behind the iris, connected to the ciliary body through suspensory ligaments [10]. The sclera is described as an extension of the cornea that consists of collagen and mucopolysaccharides [10]. The choroid is positioned between the sclera and retina which has a network of blood vessels. The retina transmits impulses to the brain via the optic nerve, thus neural function is fulfilled [1,10]. 

The ocular structure is extremely complex and specialized to provide efficient protection; thus, formulation scientists face significant challenges in delivering drugs to the eye [11]. Additionally, a wide range of pharmaceutical ingredients have negative physicochemical characteristics that restrict the therapeutic amount of drug needed to reach the desired area [11]. Consequently, the anticipated bioavailability of lipophilic and hydrophilic compounds in the anterior compartment is expected to be below 5% and 0.5%, respectively [11].

The desired concentration of drug substance at the targeted location is crucial a factor for the management of ophthalmic problems [2]. Various formulation approaches and alternative administration routes were developed to provide the required dose at the specific eye region and to escape from the restrictions of ocular barriers [1,2,10].

The tear film serves as the primary barrier for the permeability that restricts the delivery of drugs to the eyes [12]. This film is characterized as a precorneal element consisting of various layers: an external, delicate lipid layer; a central aqueous layer; and an innermost mucous layer [12]. The function of the outer lipid layer is to suppress water evaporation while minimizing the absorption of medications into both the cornea and sclera [12]. The central aqueous layer contains several proteins such as albumin and globulin [12]. These can bind and metabolize the administered medications. The innermost mucous layer has a complex blend of water, mucins, lipids, enzymes, salts, and other components [12]. The mucus layer reaches its maximum density at the epithelial apex and gradually thins out as it extends into the tear fluid [12]. The role of this layer is vital in impeding drug delivery, given its pore structure that includes negatively charged glycans and hydrophobic regions [12]. These features enable the trapping and adherence of foreign particulates. Following this, the particulates undergo elimination through the mucus turnover before reaching the corneal surface [2]. 

Nasolacrimal drainage and tear flow could be considered to be dynamic barriers [2]. The typical tear volume is around 7 µL, yet it significantly increases to 30 µL after topical application [2]. This leads to the immediate drainage of excess fluid through the nasolacrimal duct, resulting in the loss of over 85% of the applied dose [2]. Furthermore, the drug that remains may experience additional dilution due to the fast turnover of tears [3]. This process reduces the concentration gradient and diffusion rate, leading to a decreased bioavailability of intraocular drugs in the aqueous humor [3].

The cornea serves as a visible barrier against both hydrophobic and hydrophilic molecules due to its mechanical and chemical properties [13]. The cornea’s optically transparent and avascular structure comprises five well-organized layers: the epithelium, Bowman’s membrane, stroma, Descemet’s membrane, and endothelium [13]. The trans-corneal permeation of drugs is primarily limited by the epithelium and stroma [13]. The external multilayer epithelium of the cornea is hydrophobic and characterized by tight junctions among epithelial cells [13]. These tight junctions create a formidable barrier for hydrophilic drugs, as they mainly rely on paracellular channels for diffusion [13]. In contrast to the epithelium, the stroma functions as a hydrophilic fibrous layer, restricting the entry of hydrophobic molecules that passively move through the transcellular pathway [1]. The permeation from the cornea is significantly affected by both the molecular size and the hydrophobic or hydrophilic characteristics of the drug [1]. 

The ciliary body is positioned behind the iris and produces aqueous humor, which provides nutrition to the internal structures of the eye [14]. The produced aqueous humor moves toward the cornea, collects within the Schlemm’s canal, and empties into the episcleral blood vessels [14]. Consequently, the aqueous humor acts as a dynamic barrier to removing therapeutic drugs from the ocular structures [14]. The rapid turnover of aqueous humor further diminishes the concentration of the medicines, exacerbated by resistance to the direction of aqueous humor flow [14]. As drugs gradually migrate from the aqueous humor to the posterior compartment, they contact the iris and lens. Knowledge from the literature indicates that the ciliary body and iris express specific active drug transporters, thereby hindering the permeation of drugs [14]. Moreover, the melanin pigment in the ciliary body attaches to the drug, and the iris can obstruct the drug from reaching the posterior compartment [14]. 

Alternatively, topically applied drugs can access the conjunctiva and enter the eye through the conjunctiva–sclera–choroid route [2]. The conjunctiva is a glassy mucous membrane surrounding the eye’s surface near the cornea, featuring an upper epithelium and a lower stromal layer with rich blood and lymphatic vessels [2]. Despite this, due to its rich vascularization, drugs that permeate the conjunctiva might enter the systemic bloodstream through the conjunctival sac or nasal cavity instead of reaching the ocular regions [15]. Hence, this may result in a substantial loss of drugs, particularly affecting smaller hydrophilic molecules and, consequently, diminishing their ocular bioavailability [3].

The sclera poses a major challenge in ophthalmic drug delivery with its limited permeability [14]. The sclera comprises a dense, hydrophilic, collagenous connective tissue that makes up the outer shell of the eyeball [14]. This structure is characterized by an overlapping scleral collagen matrix, along with a negatively charged proteoglycan matrix in the inter-fibril space [16]. Transscleral drug delivery is significantly impacted by the thickness of the sclera [2,16].

The choroid, positioned between the sclera and the retina, constitutes a notable dynamic barrier due to its rich vascularization and innervation, which supply blood to the retina [17,18]. The choroid comprises five layers arranged from the exterior to the interior: the suprachoroidal cavity, two vascular layers, the choroid capillary layer, and Bruch’s membrane [17,18]. Bruch’s membrane, the basement membrane for the retinal pigment epithelium (RPE), is a thin collagenous membrane between the choriocapillaris externally and the RPE internally [17]. Given that the RPE expresses various enzymes (esterases, peptidases, dehydrogenases, and cytochrome P-450 enzymes) and efflux proteins (P-gp), it acts as a metabolic barrier impeding drug permeability [17]. The Bruch’s–choroid complex constitutes a more substantial obstacle to drug delivery through the transscleral pathway than the sclera alone [17]. It can bind solutes, particularly positively lipophilic drugs, resulting in the formation of a slow-release drug depot within the Bruch’s–choroid complex [17]. Additionally, molecular size plays a role in the permeability of the Bruch’s–choroid complex, with hydrophilic carboxyfluorescein and dextrans exhibiting an exponential decrease in permeability as their molecular radius increases in bovine tissues [17].

Following intravitreal administration, the vitreous humor acts as the initial barrier, preventing drug permeation into the underlying retinal and choroidal tissues [19]. This intraocular delivery method is extensively used to establish drug concentrations in the posterior eye [19]. The viscosity of the vitreous fluid hinders the diffusion of larger and heavier therapeutic cargoes, with minimal impact on the diffusion of smaller molecules [19]. Furthermore, the net anionic charge of the vitreous humor regulates the diffusion of drug molecules [15]. Negatively charged particles can diffuse freely, while positively charged particles become entrapped in the vitreous medium [15]. Therefore, the molecular weight and charge of the administered drug significantly influence its distribution within the vitreous and its bioavailability in the retina [15]. 

The blood–ocular barrier (BOB) system includes the anterior blood–aqueous barrier (BAB) and the posterior blood–retinal barrier (BRB) [20]. The BAB, found in the anterior compartment of the eye, consists of vascular endothelium without pores covering the iris blood vessels and tight junctions linking the apical portions of neighboring epithelial cells of the non-pigmented ciliary body epithelium [20]. This structure prevents drug entry from the plasma into the aqueous humor [20]. Nevertheless, the BAB is still not an absolute barrier due to the porous capillaries in the ciliary body stroma [20]. These permeable capillaries, serving as a secondary source of plasma protein leakage to the iris, also facilitate the entry of small molecules into the iridial circulation [20]. The BRB primarily hinders drug diffusion into the retina after systemic circulation [20]. The BRB found in the posterior compartment can be further categorized into the inner and outer BRB [20]. The inner BRB, composed of retinal capillary endothelial (RCE) cells, features tight intercellular junctions that selectively shield the retina from foreign substances in the bloodstream, especially hydrophilic compounds and macromolecules [20]. The sturdy intercellular junctions within the retinal microvasculature are robust structural barriers, effectively preventing molecular diffusion to and from the retina [21]. This is attributed to the characteristics of the retinal microvascular endothelium, which lacks permeable openings and expresses specialized intercellular junction proteins [21]. On the other hand, the outer BRB, which separates the choroid and Bruch’s membrane from the inner retina, is established by the tight junctions between RPE cells [22]. This architectural arrangement restricts the entry of administered drugs to the retina and vitreous region to only 1–2% [21,22].

## 3. Ocular Drug Administration Routes

Achieving an effective drug concentration at the lesion site is crucial for managing ocular diseases [22]. Various administration routes, such as systemic, topical, intraocular, and periocular administration, have been developed to attain effective drug concentrations at target locations while navigating existing ocular barriers [22,23].

Effective delivery of medications to ocular tissues through systemic administration, including intravenous and oral dosing, encounters notable challenges for various reasons [22]. Firstly, when drugs are administered orally, they may be affected by the physiological conditions of the gastrointestinal tract and undergo a first-pass effect [22]. Secondly, the eye’s limited blood supply results in minimal drug accumulation [22]. Most importantly, the BOB acts as a formidable obstacle, preventing drugs from reaching ocular tissues [22]. The anterior BAB restricts drug penetration into the eye’s anterior segment from the systemic circulation, while both the inner and outer BRB allow only restricted drug administration in the posterior eye compartment [22]. This necessitates high-dose and frequent drug administration to achieve therapeutic effectiveness, potentially leading to systemic side effects and challenges in therapy compliance [18].

Topical administration remains the favored approach for addressing ophthalmic disorders, owing to its non-invasive delivery method [23]. Despite its widespread use, topical administration faces a persistent challenge in achieving optimal drug delivery efficiency [23]. The primary concern continues to be pre-corneal drug loss, largely influenced by the rapid turnover of tears, frequent blinking, lacrimation, and nasolacrimal drainage [23]. As a result, there is a necessity for frequent instillation of eye drops to maintain the necessary drug concentration on the ocular surface, potentially resulting in suboptimal compliance and complications [23]. Moreover, treating posterior eye disorders with topical application proves less effective, even with repeated dosages, due to the persistent presence of the anatomical corneal barrier [23]. At the same time, it is crucial not to overlook the repercussions of side effects, including damage from preservatives, ocular irritation, complications stemming from steroid use, and others [19]. 

In intraocular administration, key routes include intracameral, intravitreal, subretinal, intrastromal, suprachoroidal, and intrastromal options [24]. In intracameral injection, the drug is directly injected into the anterior chamber, a practice commonly used post-cataract surgery and for the therapeutic management of anterior compartment diseases, including fungal and bacterial keratitis [24]. Nevertheless, this approach proves ineffective in transporting drugs to the posterior segment due to the challenge of drug penetration against the flow of aqueous humor in the eye [24]. In consideration of this, IVT is embraced to establish drug concentrations in the posterior portion of the eye [24]. This method includes both direct IVT and the incorporation of implantable devices, establishing itself as the leading approach in managing vitreoretinal diseases in recent decades [24]. It enables the precise delivery of therapeutic agents, such as anti-vascular endothelial growth factor (VEGF) [25], steroids [26], and genes [27] resulting in elevated drug concentrations in both the vitreous and the retina [24]. Despite this, there is a significant risk of ocular complications, including bleeding, retinal holes, cataracts, vitreous hemorrhage, elevated intraocular eye pressure, secondary glaucoma, optic nerve injury, endophthalmitis, and retinal toxicity [28]. Subretinal injection is implemented to effectively access and distribute agents within the subretinal space, situated between RPE cells and photoreceptors [28]. This targeted delivery method seeks to improve the treatment outcomes for vision diseases caused by mutations in photoreceptors and/or RPE genes, in addition to various retinal degenerative diseases [28]. Suprachoroidal administration is applied for implants, microneedles, and other formulations, directing the drug between the sclera and choroid to reduce systemic side effects [29,30,31]. Meanwhile, intrastromal injection, known for its minimally invasive approach, enhances drug delivery to the affected corneal stroma without compromising tissue structures [32].

Periocular administrations offer a prolonged duration of action compared to IVT, attributed to the capacity to inject larger volumes [33]. Additionally, there is a relatively low risk of ocular pain, infection, endophthalmitis, or bleeding [33,34]. SCT, the predominant form of periocular administration, is positioned between the bulbar conjunctiva and sclera [34]. This method is frequently employed in clinical practice to administer drugs, including local anesthetics and anti-inflammatory medications, specifically targeting the anterior segment of the eye [34]. Despite its common use, subconjunctival medications often encounter limitations in bioavailability as they are highly absorbed by the lymphatic and blood circulatory systems rather than achieving optimal intraocular distribution [34]. Consequently, frequent injections are necessary, introducing operational risks such as conjunctival edema and subconjunctival hemorrhage [34]. The remaining periocular application methods, including posterior juxta scleral, retrobulbar, peribulbar, and sub-tenon applications, are primarily employed for anesthesia during ocular surgery [35]. It is noteworthy, however, that periocular administration still struggles to deliver an adequate amount of drugs to the retina due to losses occurring in the periocular space, the BRB, and the choroidal circulation, among other factors [33,34,35]. 

## 4. Vesicular Systems

Site specificity, improved bioavailability, and stability are just a few of the objectives that lipid particle systems have been developed to accomplish [36]. Vesicular drug delivery systems like liposome, niosome, ethosome, transfersome, and others (bilosome, transethosome, cubosome, proniosome, chitosome, terpesome, phytosome, discome, and spanlastics) can encapsulate both hydrophilic and lipophilic drugs [3,4,5,6]. The aqueous core and lipid bilayer, both structural components of vesicles, carry both polar and non-polar drugs during delivery (Figure 1). Drugs embedded in lipid vesicles can easily pass through cell membranes, which alters both the rate and scope of drug absorption as well as drug distribution. The vesicular systems have been successfully developed to impart a variety of qualities, such as (a) a prolonged half-life in the body, (b) targeting to the site of the disease, (c) high trapping; (d) a minimum amount of side effects; and (e) enhanced bioavailability [37]. Based on their main components, vesicular delivery systems can be further divided into various groups (Figure 2).

For the ocular route, vesicular systems enable controlled ocular delivery by preventing the drug’s metabolism by the enzymes at the tear/corneal epithelium interface and supporting sustained and controlled action at the corneal surface [6]. Also, vesicles present a potentially effective way to meet the need for an ocular drug delivery system that has the availability of a drop but will localize and maintain drug activity at its site of action [6]. In addition to the physicochemical characteristics of the drug itself, such as its solubility and particle size in the case of suspensions, the rate of drug penetration also depends on the characteristics of its carrier [6]. 

The ocular use of vesicular systems has many advantages as we have mentioned. In addition to all these advantages, there are disadvantages such as oxidation and hydrolysis-like reactions, low solubility, leakage and fusion of encapsulated drug/molecules, stability, high production cost, and short half-life [38]. Intraocular clouding may occur with administration via intravitreal injection [38]. Liposomal formulations generally require more complicated clinical trials and an expensive manufacturing process than conventional formulations [38]. In addition to general difficulties including short shelf life, low drug loading, and sterilization issues, liposomes face ocular-specific difficulties such as limited retention time for intravitreally injected liposomes and unclear ocular safety during prolonged and repeated use [39]. These obstacles need to be cleared before liposomes are applied topically [39]. For niosomes, insufficient clinical data are available for surfactant-related in vivo toxicity and irritation. Another vesicular systems, cubosomal formulations, are challenging to scale up and manufacture on large scales because of their complicated phase behavior and viscosity-related complexity [3]. 

Nanoparticulate drug delivery systems provide the capability for delivering therapeutics to specific ocular targets [39]. The use of polymeric systems (nano-emulsion, dendrimer, polymeric micelle, solid lipid nanoparticle, polymeric nanoparticle, micro- and nano-spheres) was investigated for ocular drug delivery [39]. Although current research shows that nanoparticulate delivery methods have immense therapeutic potential, transferring these systems from bench to bedside represents a difficult challenge [39]. For dendrimers, only a few clinical trials have been launched, and no safety or tolerability outcome has been announced [39]. Particle growth, unpredictable gelation tendency, and unexpected polymorphic transition dynamics are some stability issues of solid lipid nanoparticles and polymeric nanoparticles. Long-term and repeated use of nanoparticle systems requires further research [39]. Among these carriers, liposomes have been the most studied [40]. Liposome includes an aqueous core entrapped by one or more bilayers composed of natural or synthetic lipids, which makes them biodegradable and biocompatible [40]. They are composed of natural phospholipids that are biologically inert and they have low toxicity [40]. Drugs with different lipophilicities can be encapsulated into liposomes: strongly lipophilic drugs are entrapped in the lipid bilayer and hydrophilic drugs are located in the aqueous core [40]. 

### 4.1. Liposome

Liposomes were first offered by Alec Bangham in 1965 [41]. Liposomes are bilayer vesicles made of phospholipids that typically have vesicle diameters between 0.08 and 10.00 μm [38,42]. According to established classification, they are divided into three categories based on their vesicle size: small unilamellar vesicles (SUVs) with a size range between 10 and 100 nm; large unilamellar vesicles (LUVs) with a size range between 100 and 300 nm; and multi-lamellar vesicles (MLVs) with a size range of >500 nm [6]. Drugs such as anticancer, antifungal, antiviral, local anesthetics, and biomolecules, which include vitamins and enzymes, have all been given liposomal forms [43]. Their similarity to normal cells makes them effective carrier systems [43]. Liposomes have low toxicity [43]. Targeting can be accomplished by attaching moieties to the lipid membrane [43]. 

Since liposomes can contain both hydrophilic and lipophilic molecules and have high compatibility with ocular surfaces, they are an efficient ocular drug carrier [44,45]. In numerous studies, liposomal ocular delivery has been demonstrated to be effective for both the anterior and posterior segments of the eye [3,4,5,6]. Liposomal technology successfully facilitates sustained/controlled or modified drug release and also improves bioavailability and biocompatibility [3]. They can reduce the toxicity of certain drugs to the eyes [3]. Liposomal systems have several disadvantages that are short half-life, storage difficulties, poor hydrophilic drug entrapment efficiency, and blurred vision after IVT [3,9]. Liposomal ocular delivery is highly biocompatible with tear film due to its structural features [3]. Liposomes special physicochemical properties make them effective ocular carriers because they limit drug clearance, have excellent stability, and increase transcorneal uptake [46]. Liposomes are widely studied in ocular drug delivery [3,4,5,6]. For this reason, only studies from the last 5 years have been evaluated for liposomes in this review. These studies are also summarized in Table 1 at the end of the chapter title.

Qiao et al. (2022) designed rebamipide (RBM) liposomal eye drops for the treatment of dry eye disease that have excellent stability, good patient compliance, and a slow release rate [47]. These eye drops are prepared by remote loading technique and optimized with various parameters. Different formulations were prepared by using varied hydrogenated soybean phospholipid-to-cholesterol (HSPC-to-Chol) ratios (2.5, 3.5, and 4.5), drug-to-lipid ratios (0.2, 0.1, and 0.05), calcium acetate solution concentrations (100, 200, 300 mM), incubation times (10, 30, 60, 90 min), and temperatures (60, 65, 70 °C). The encapsulation efficacy and in vitro release of RBM were evaluated. Compared with RBM suspension, RBM liposomes have a slower release rate and less cumulative release. Liposomal RBM showed enhanced retention time at the corneal surface and improved drug penetration. 

In one study, Campardelli et al. (2018) used a supercritical CO_2_-based one-step continuous process, named supercritical assisted liposome formation (SuperLip), for the production of ampicillin and ofloxacin-loaded liposomes for opthalmic drug delivery [48]. The results showed that ofloxacin and ampicillin exhibited encapsulation efficiencies of up to 97% and 99%, respectively. The study reveals that the important factor to improve encapsulation efficiency was found to be the alteration of the lipid-to-water ratio. A rise in this ratio permitted the formation of vesicles. Liposome stability lasted for at least three months. 

In a study conducted by Moustafa et al. (2017), hyalugel-integrated liposomes (HYS7) were designed to enhance corneal permeability for fungal keratitis treatment [49]. Liposomes were prepared with a simple TFH technique and were investigated in a series of different formulations. In vitro optimization was accomplished concerning hyaluronic acid (HA) and fluconazole (FLZ) concentration, entrapment efficiency, particle size, and stability. When compared to the FLZ solution and conventional liposomes, FLZ in HA gel and the chosen HYS7 demonstrated a considerable delay in the initial drug release and a decrease in the dosing frequency. After 6 h from HYS7, ex vivo cumulative FLZ corneal penetration was 2.99 and 4.18 folds greater than FLZ suspension and conventional liposomes, respectively. In vivo corneal penetration of HYS7 demonstrated a remarkable sustained effect of FLZ.

As demonstrated by Dong et al. (2015), utilizing a mucoadhesive substance is an effective way to increase ocular drug therapeutic efficacy [8]. The goal of this study was to create a liposomal formulation coated with silk fibroin (SF), a new adhesive excipient, for topical ocular drug delivery. Liposomes containing ibuprofen were prepared by the ethanol injection method. The ibuprofen-loaded liposomes were coated with the regenerated SF that has various dissolving times. The morphology, drug encapsulation efficiency, in vitro release, and in vitro corneal permeation of SF-coated liposomes (SLs) were studied and contrasted with the conventional liposome. To examine the impact of SF coating on the drug release behavior, in vitro release tests were performed using the solution, liposomes, and SLs containing ibuprofen. According to the results, an increased release rate was observed especially in SLs. In comparison to liposomal samples without coating, SLs have more sustained drug release behavior, and the release rates are reduced as the SF concentration increases. 

Tan et al. (2017) developed timolol maleate (TML) bioadhesive chitosan-coated liposomes (TM-CHL) for the treatment of glaucoma. TML-containing liposomes are prepared by ammonium sulfate gradient coupled with a pH-gradient method [9]. The resulting TML-CHL was the most promising formulation, having an encapsulation efficiency (%) (EE%) of 75.83 1.61% and a mean particle size of 150.7 nm. According to an in vitro release study, in comparison to TML liposomes, TML-CHL had a superior prolonged release because of the chitosan (CH) coating. To increase patient compliance, the frequency of administration was able to be reduced. In vitro transcorneal permeation tests show that the cumulative amount of drugs from TML-CHL was higher than TML liposomes due to the addition of CH increasing transport. Ocular pharmacokinetic study indicates that TML-CHL is retained in the precorneal region longer than TML liposomes or TM eye drops. TML-CHL exhibited a significant increase in AUC (0–∞), bioavailability, mean residence time (MRT), and Cmax. Consequently, the findings showed that CH coating enabled sustained retention in the precornea, providing a sustained action to increase drug permeability and bioavailability.

In one study, Karumanchi et al. (2018) developed Bevacizumab-loaded liposomes by modified thin-film hydration (TFH) and extrusion methods [50]. The goal of this study was to create extended released drug delivery to treat ocular angiogenesis. Different phospholipids and cholesterol were used in the preparation of the liposomes. For stable formulations, a 25–30% molar ratio of cholesterol with a 70–75% molar ratio of total phospholipid content is required. The liposomes with high concentrations of phosphatidylcholine (PC) had an optimum size of 100–200 nm, while those with low concentrations of phosphatidylglycerol increased stability and shelf life. Three formulations were prepared for encapsulating Bevacizumab–DPPC–DPPE–DPPG–cholesterol compositions of 60:10:0:30, 65:5:5:25, and 60:5:5:30. Using these compositions, conventional and stealth liposomes were obtained. A combination of various compositions would be excellent to offer extended-release delivery of drugs and to maintain the minimal effective concentration in the subject’s vitreous, based on the release profiles from various formulations. Researchers were able to demonstrate the in vitro effectiveness of the liposomes loaded with bevacizumab in terms of slow release and sustained anti-VEGF activity according to in vivo studies. 

Chetoni et al. (2015) developed a topical controlled-release liposomal formulation containing Distamycin A (DA) for the treatment of acyclovir-resistant Herpes simplex virus (HSV) keratitis [51]. This new liposomal formulation can reduce the toxicity of active ingredients and enhance drug uptake. The reverse phase evaporation technique (REV) was used to produce a liposomal formulation containing DA. The EE% was 34.53%. Fast diffusion of DA from the aqueous reference solution (DA-Sol) and a more controlled release rate from DA-Lipo vesicles were both seen in the study of the drug release. For both formulations, the total amount of DA delivered by reconstituted rabbit corneal epithelial membranes had a similar time-course profile and lag times. Both DA formulations recovered a similar amount of DA into the reconstituted corneal tissue, which was 3.40 and 4.58% of the administered dose for DA-Sol and DA-Lipo, respectively. The relevant pharmacokinetic parameters indicate that less rapid elimination of DA after the instillation of the liposomal formulation. The DA half-life in tear fluid, which was 1.82 and 2.75 min for DA-Sol and DA-Lipo, respectively, confirmed this result. The AUC findings demonstrate that the DA-Lipo formulation enhanced the bioavailability of DA in tear fluid.

Table 1 summarizes the studies conducted on ophthalmic liposomes in recent years. During these studies, production methodologies such as the remote loading technique, supercritical assisted liposome formation, TFH, and the ethanol injection method were utilized. Percentage changes in drug-loading capacity due to production methodologies were observed to range between 34% and 99% [48,51]. Although drug and excipient selections influence the EE percentage, it was observed that the supercritical-assisted liposome formation technique provided the highest EE value [48]. With the increasing use of this method, it is believed that liposomes with high drug-loading capacity can be produced more effectively [48]. When examining studies conducted in recent years, surface modification studies also stand out. The purpose of surface modification is to prolong residence time on the surface. It has been shown that modified liposomes such as stealth liposomes [44] and silk fibroin-modified liposomes [8] provide delayed or extended release. With surface modification, it is anticipated that the frequency of dosing can be reduced by developing long-acting ophthalmic liposomes.

### 4.2. Niosome

By hydrating non-ionic surfactant, cholesterol, or other amphiphilic compounds, niosomes are self-assembled vesicular nano-carriers that act as an adaptable drug delivery system with a range of uses such as oral, ocular, topical, pulmonary, parental, transmucosal drug delivery, and cosmetic applications [63]. Niosomes may include both hydrophilic and lipophilic molecules like liposomes. Additionally, niosomes have the advantages of liposomes since they can be produced using simple techniques, at a lower cost, and with longer-lasting stability. Niosomes can eliminate difficulties with large-scale manufacturing, sterility, and physical stability [64,65]. Niosomes can be prepared as unilamellar or multilamellar vesicles [66,67]. Several types of niosomes have been described in the literature. These are divided into many groups based on their size or the number of lamellar layers. SUVs and LUVs are categorized according to size. Based on the number of bilayers, there are MLVs and SUVs [68]. 

Niosomes have been investigated as a potential ocular drug delivery system for many therapeutic drug classes. They can offer many advantages over other ocular drug delivery systems [6]. They have reduced production costs, are made of biodegradable and nonimmunogenic materials, and are chemically more stable than liposomes [6]. Niosomes do not need to be handled in an expensive manner (stored in freezers and prepared under nitrogen gas), unlike phospholipids [6]. With enhanced physical stability and targeted distribution at the site of action, they enhance drug performance [6]. Niosomes are studied for the ocular delivery of drugs because of their low toxicity and penetration-improving properties [4]. Numerous colloidal drug delivery systems, including niosomes, have been developed and used as potential ocular delivery platforms to cope with the disadvantages of the conventional ophthalmic dosage forms, such as insufficient residence time, loss of drug via nasolacrimal drainage, inadequate corneal penetration, which together result in low bioavailability and the need for frequent instillations [4,5]. 

Baldino and Reverchon (2022) prepared nano-niosomes by a continuous supercritical CO_2_ assisted process [69]. These liposomes are used for the treatment of bacterial conjunctivitis. For the optimum formulation, 145 ± 52 nm in diameter, 90/10 Span 80 to Tween 80, and at a surfactant-to-cholesterol molar ratio equal to 2 and 4 is used. The vesicle’s mean diameter generally increased slightly as a result of the cholesterol added to the niosomal formulations but ofloxacin encapsulation efficiency and release time (i.e., 78% EE% and 5 h release time) showed the most significant effects. In a previous study, the method used for niosome production, namely SuperLip, was designed [70]. However, because this is an adaptive lab-scale plant that can also be utilized for other types of vesicles, we decided to call it SuperSomes. Three feeding lines connect to form the high-pressure plant known as SuperSomes. 

A study conducted by Gugleva et al. (2019) prepared and characterized niosomes for the ocular delivery of doxycycline hyclate [71]. The encapsulation of doxycycline into niosomes as a potential drug delivery system offers a potential solution for local therapeutic management of MMP-mediated ocular surface diseases. TFH method followed by multiple membrane extrusion or the REV was used to generate niosomes utilizing a variety of surfactants (Span 20, Span 60, Span 80, and Tween 60) and cholesterol in varying molar ratios. Niosomes prepared by the REV method have higher encapsulation efficiency in comparison to niosomes prepared by the TFH extrusion method. The formulation made up of Span 60 and cholesterol and manufactured using the REV demonstrated the maximum entrapment efficiency (%EE 59%). In vitro release studies showed sustained release of doxycycline from niosomes. According to the obtained release profiles, all of the evaluated niosomal formulations tend to release the encapsulated doxycycline over a period of up to 20 h.

Naltrexone (NTX) niosome was used for the treatment of diabetic keratopathy and produced using the REV [72]. Abdelkader et al. (2012) prepared NTX niosome for evaluation of conjunctival and corneal tolerability on the hen’s egg chorioallantoic membrane [73]. For this purpose, the hen’s egg test chorioallantoic membrane (HET-CAM), bovine corneal opacity, and permeability (BCOP) test were used. As the HET-CAM test responds to irritant compounds with an inflammatory reaction similar to that produced by conjunctival tissue, it can be used as an acceptable model for conjunctival irritation testing [74]. The irritant potential of all the generated niosomal formulations (total lipid concentrations up to 10%, *w*/*v*) was evaluated, and the results revealed no signs of inflammatory reactions. Similar results have been reported on niosomes assessed using the in vivo rabbit eye test (Draize test) [5]. The findings of the HET-CAM test were well associated with the test substances’ assessments for corneal opacity and fluorescein permeability. The HET-CAM and BCOP results were found to agree with the findings on corneal erosion and stromal thickness.

El-Haddad et al. (2021) developed a resveratrol-loaded chitosan-coated niosomes (chitoniosomes) by ethanol injection method [75]. Because of the surface coating, chitoniosomes show increased efficacy and sustained-release profile. Resveratrol-loaded niosomes are used for the treatment of ocular inflammation. The findings revealed that the chitoniosomes (RSV.CsNiPolx) had a 1.9 times greater mucoadhesive effectiveness than the uncoated niosomes and showed good stability in the refrigerator (4 °C) over 6 months. The in vitro release profile showed that resveratrol-loaded chitoniosomes decreased the release rate. The CH layer can soak up water and expand, forming hydrating channels that enable sustained drug delivery. In vivo animal study tests showed that niosomes formulation has good in vivo ocular tolerance and safety. The inflammatory mediators TNF and IL-6 were reduced in semi-quantitative PCR, which is evidence of the in vivo anti-inflammatory effects.

Allam et al. (2021) developed niosomes that incorporated into pH-responsive in situ forming gels to enhance the ocular availability of betaxolol hydrochloride for management of glaucoma [76]. They are produced by the TFH technique followed by sonication. The optimized niosomes, composed of Span^®^ 40 and cholesterol at a molar ratio of 4:1, displayed a particle size of 332 ± 7 nm, a zeta potential of −46 ± 1 mV, and an encapsulation efficiency of 69 ± 5%. 

In studies regarding niosomes, various production methodologies have been identified, including TFH, the continuous supercritical CO_2_ process, reverse-phase evaporation, and ethanol injection [5,69,70,71,72,73,74,75,76]. The resulting niosomes from these methods have shown EE percentages ranging from 40% to 90% [75]. While both drug and excipient selections influence the EE percentage, it was noted that the ethanol injection method yielded the highest EE value [75]. In this regard, the ethanol injection method has been evaluated as a highly efficient technique for niosomal systems in terms of EE. The increasing number of studies will further solidify the results of this method. In addition, it can be noted that a distinctive evaluation compared to ophthalmic liposomal systems is the achievement of stability for up to 6 months in the medium to long term [75]. 

### 4.3. Ethosome/Transethosome

With a strategy focused on modifying vesicle compositions, ethosomes (deformable lipid-based elastic vesicles) have been developed as systems capable of delivering active components to underlying tissues [77]. Due to their smaller size and elastic shape compared to typical liposomes, they enable local transport of both hydrophilic and hydrophobic active molecules and can enhance drug penetration or localization [78,79]. The comparative study revealed that ethosomal formulations exhibited more drug penetration into the cornea than liposomal formulations [80]. 

Uner et al. (2023) developed TML-loaded ethosomes for the treatment of glaucoma by ophthalmic drug delivery [77]. The TFH method and Box–Behnken experimental strategy were used for the preparation and optimization of ethosome formulations. The composition of formulations consisted of TML (5 mg/mL), SPC (3–5%), and poloxamer188 (P188) (2–6%). The particle size, zeta potential, and encapsulation efficiency (EE %) were found to be 88.23 ± 1.25 nm, −28.7 ± 2.03 mV, and 89.73 ± 0.42%, respectively. An ex vivo permeation study showed that TML-loaded ethosomes could deliver TML to the cornea in an efficient manner by gradually releasing TML throughout the corneal tissue. For the irritation assay, the HET-CAM method was used for the safety assessment. The results revealed that the TML-ET-1 formulation is nonirritant. According to an in vivo study, the new drug delivery system demonstrated similar results to conventional eye drops.

In research, Sushma et al. (2023) designed ethosomes (IAEs) for a combination photothermal therapy of fungal keratitis that contains the near-infrared (NIR) dye, Indocyanine green (ICG), and the anti-fungal drug Amphotericin B [80]. Ethosomes were prepared by the TFH method. Compared to a single therapy, the IAEs demonstrated synergistic in vitro cytotoxicity against C.albicans. IAEs are biocompatible both in vivo and in vitro, and they may also be effective in photothermal transduction. In ex vivo permeation tests using goat cornea, it was shown that the IAEs improved penetration. In vivo tests demonstrate the excellent photothermal properties of IAEs and their role in the treatment of fungal keratitis, which together can enhance the therapeutic effects of amphotericin B and other conventional anti-fungal drugs. 

Trans-ethosomes, one of the lipid-based nanovesicles, include an edge activator and ethanol [81]. Trans-ethosomes involve the advantages of both ethosomes and transfersomes. Ethanol improves drug penetration through tiny passageways created in the stratum corneum as a result of fluidization by increasing the fluidity of lipids and decreasing the density of the lipid bilayer [82]. The phospholipid bilayer is weakened by the edge activator, which also increases the vesicle’s elasticity and deformability [83]. Trans-ethosomes are superior to other delivery systems in terms of drug distribution because of the incorporation of these components. 

Ahmed et al. (2021) formulated ketoconazole (KET) trans-ethosomal vesicles loaded into different ophthalmic in situ gel (ISG) and hydrogel to increase KET ocular permeation, antifungal activity, rapid drug drainage, and short-elimination half-life [81]. KET trans-ethosomes were prepared by the TFH method. The results revealed that the drug-to-phospholipid-molar ratio, the percentage of edge activator, the percentage of ethanol, and the percentage of stearyl amine significantly affect the characteristics of the vesicles. The optimized vesicles were spherical and showed an average size of 151.34 ± 8.73 nm, a zeta potential value of +34.82 ± 2.64 mV, an entrapment efficiency of 94.97 ± 5.41%, and a flexibility of 95.44 ± 4.33%. It was discovered that the produced ISG compositions did not irritate the cornea. Without having any harmful effects, the trans-ethosomes vesicles can penetrate deeper into the posterior eye segment. 

In studies conducted on Ethosomes/Transethosomes up to the present, it has been determined that the TFH method is the production method utilized [77,78,79,80,81]. Interestingly, the ophthalmic Ethosomal/Transethosomal carrier systems prepared using the TFH method have demonstrated entrapment efficiencies ranging between 80% and 90%, a notable contrast to ocular liposomal and niosomal systems [77,81]. This is believed to be due to the alcoholic content serving as a formulation component.

### 4.4. Transfersome

Transfersomes, ultra-flexible deformable nano-liposomes, are prepared by adding surfactants to the phospholipid bilayers of conventional liposomes [84]. Since they contain phospholipids, they are biodegradable and biocompatible nanocarriers [85]. They might prevent the encapsulated drugs from degrading through the metabolism and have great encapsulation efficiency. The edge activators increase the flexibility and deformability of the transfersomal vesicles by acting as destabilizing features in the vesicular membrane. Transfersome are used in ocular drug delivery for enhanced encapsulation efficiency. 

The main difficulties with topical ocular drug delivery are short precorneal residence time and inadequate transocular membrane permeability. In research conducted by Janga et al. (2019), the effectiveness of the natamycin (NT) transfersomes electrolyte-triggered sol-to-gel-forming mechanism was studied for improved and sustained ocular delivery [86]. The NT transfersome formulations (FNs) were prepared by the TFH technique and optimized by different molar ratios of a phospholipid, sorbitan monostearate (Span), and tocopheryl polyethylene glycol succinate (TPGS). Formulations loaded with gellan gum (0.3% *w/v*) (FNGs) instantly generated an ISG with significant viscoelastic properties. In vitro cytotoxicity in corneal epithelial cells and corneal histology studies showed that ocular safety of optimized formulations. The transcorneal permeability of NT was significantly higher than in the control suspension. In vivo test results demonstrate the potential of transfersome-loaded electrolyte-responsive sol-to-gel–transforming systems as a reliable system for sustained ocular drug delivery. 

Acetazolamide (ACZ)-loaded transgelosomes (TGS) are developed for improving corneal delivery [85]. ISG-forming solutions provide prolonged drug release and higher bioavailability. ACZ-loaded transfersomes were prepared by the ethanol injection method, using PC and different edge activators, including Tween 80, Span 60, and Cremophor RH 40. Using poloxamers functioning as the gelling agents, the optimal formula has been developed as TGS. TGS demonstrated longer drug release (71.28 ± 0.46% after 8 h) and higher ex vivo permeation (66.82 ± 1.11% after 8 h), as well as a significant reduction in intraocular pressure (IOP) over 24 h, as compared to free drugs.

### 4.5. Other Vesicular Systems

Recently, different kinds of vesicular systems are investigated for ocular drug delivery such as bilosome [87,88,89], cubosome [90,91,92], proniosome [93,94,95,96], chitosome [97,98], terpesome [99,100], phytosome [101], discome [102], spanlastics [103,104], flexosome [105], phytocubosome [106], and oleophytocubosome [107]. These systems have more advantages than conventional vesicular systems, for instance, enhanced corneal permeability and elasticity, and improved tear/corneal surface contact time. Bilosomes, closed bilayer vesicles of non-ionic amphiphiles, contain bile salts and are closely related to niosomes. The presence of bile salts and surfactants, as well as the nano-size of bilosomal vesicles, indicates great possibilities for evaluating them via the ocular route [87]. Cubosomes are identified by their distinctive and thermodynamically stable isotropic structure, which is made up of a bicontinuous curved lipid bilayer in three dimensions and two in-line water channel networks [108,109]. They are excellent candidates for pharmaceutical delivery because of their low viscosity, great heat stability, large surface area, and ability to encapsulate both hydrophilic and hydrophobic medicines [110]. They have a similar structure to biological membranes.

Proniosomal gels are vesicular systems with nonionic-based surfactants, cholesterol, and other substances in the vesicles. Upon hydration, these vesicles can transform into niosomal suspensions [95]. CH, natural, linear, and cationic polysaccharide polymers are substances commonly used to produce ophthalmic delivery systems [97]. Niosomes are coated with CH, which increases the retention time as a result of mucoadhesion action and improves penetration [111,112]. Terpesomes (TPs) are terpenes containing vesicles which are naturally occurring substances derived from essential oils and made up of several isoprene units first described by Albash et al., for the treatment of ocular candidiasis [99]. Due to their residence in the lipophilic components of the cell bilayer, terpenes are not only well-known penetration enhancers but also have antimicrobial and antifungal properties. This enables the delivery of essential oil constituents to the interior of the cell, where they cause cytoplasmic infiltration and cell death [113]. 

Phytosomes are supramolecular complexes that are compatible with lipid drugs and self-assemble into vesicular structures that resemble liposomes but have a different localization for the guest [114]. Water-soluble drug molecules are trapped in the aqueous core of liposomes in contrast, drug molecules in phytosomes are electrostatically aligned and hydrogen-bound to the polar head groups of phospholipids [101]. In comparison to liposomes, phytosomes have a higher drug-to-carrier load and they are also smaller and more biocompatible, making them preferred carriers for hydrophilic drug molecules [115]. One of the non-conventional forms of niosomes is called discomes, formed with the incorporation of poly-24-oxyethylene cholesteryl ether, or so-called Solulan C24 and they are enormous (approximately 20 µm in diameter), disc-shaped niosomes that coexist with more typical spherical niosomes (2–5 µm in diameter) [72]. Spanlastics (SPs) are highly elastic novel surfactant-based nanovesicles that consist of an edge activator (EA) and a nonionic surfactant [116]. They have high chemical stability, target specificity, convenience, and high patient compliance [117]. Some recently developed vesicular systems for ocular drug delivery are presented in Table 2.

## 5. Functionalization of Vesicular Systems

Despite the many advantages of lipid-based systems as mentioned in previous titles, limitations such as the instability of the chemical structure due to hydrolysis, oxidation, and leakage of the drug into the capsule limit their use [38]. Various modifications in surface properties have been investigated to obtain an ideal drug carrier. Coating liposomes with polyethylene glycol (PEG), a biocompatible and inert polymer, is one of the most commonly used surface modification methods [44]. PEGylation forms a protective layer over the liposome surface, provides a stearic hindrance to recognition by opsonins and, so, reduces their clearance [118,119]. These modified liposomes are known as ‘‘stealth liposomes’’. The stealth properties of liposomes are important for topical drug delivery targeting the posterior segment through noncorneal routes [44]. The FDA-approved non-ionic surfactant D-alpha-tocopheryl polyethylene glycol succinate (TPGS), one of the functional materials used to modify the liposome surface, is being used more frequently because of its biocompatibility, lack of toxicity, adaptability to different administration routes, and capacity to improve solubilization, stability, penetration, and overall pharmacokinetics [120]. The TPGS-modified nanocarriers may improve drug absorption by passively increasing trans-corneal permeability of ocular drugs or by lowering P-gp-mediated efflux and enhancing drug stability [121,122]. Jin et al. developed TPGS-modified nano-liposomes to deliver brinzolamide to the eye for glaucoma therapy [123]. The TPGS-modified liposomes provided higher encapsulation efficiency (over 90%) and more sustained drug release compared to conventional liposomes and enhanced in vitro penetration of brinzolamide across rabbit cornea 2 and 5 times higher than conventional liposomes and the marketed Azopt^®^ suspension formulation, respectively [123]. Liposome surface engineered with antibody fragments and monoclonal antibodies used as targeted delivery of liposomes [44]. These are called “immunoliposomes” that have antibodies attached to their surface, which recognize specific proteins on target cells [44]. Although antibodies used in ocular angiogenesis show high specificity, bioactivity, and low toxicity, high protein concentration in the eye causes toxicity. In a study by Karumanchi et al., it was observed that bevacizumab-loaded liposomes provided prolonged drug release and increased patient compliance [50]. Thus, the drug level reaching the target area is ensured by reducing the frequency of dose administration [50]. For instance, the bile salt content of bilosomal vesicles provides extra elasticity to the vesicle, resulting in increased corneal permeation. As we mentioned in our article, silk fibroin [8] and chitosan coating [9] of liposomes are other surface modification techniques that increase the residence time and ocular drug permeability in the precorneal area. 

## 6. Future Perspectives on Scientific and Commercial

The eye has always been an interesting and challenging field for pharmaceutical technologists due to its unique anatomical and physiological structure and the limited bioavailability of conventional drugs [124]. When a search is made under the title “ocular drug delivery systems” in Pubmed, 1424 articles appear only in the last 5 years (2462 in the last 10 years) [125]. If we filter these studies as review, systematic review, meta-analysis, book or book chapters, we see that there are 502 publications (786 in the last 10 years). According to these data, we see that 922 publications published in the last 5 years consist of clinical studies, research articles, case reports, etc. These numerical data show how important shared ocular drug delivery systems are in scientific research. As we examined in this review, there have been 412 research articles on vesicular ocular drug delivery systems in the last 5 years. Since there are many different topics studied under the title of “ocular drug carrier systems” (micro and nanoparticles, micelles, micro and nanoemulsions, hydrogels, dendrimers, etc.) [126], this number shows how important vesicular drug carrier systems are in the ocular field.

When we examine commercial developments, we see that the licenses of commercial products containing vesicular ocular drug delivery systems have been approved and are now available for patient use (Table 3). None of the products actually contain APIs. All of them support tears and are used to improve or alleviate the symptoms of dry eye disease. Another striking point about these products is that the carrier system of all of them is liposome [127,128,129,130,131,132,133,134]. This may be due to several reasons: 

Liposomes are the first group developed among vesicular systems. For this reason, their knowledge is greater than other vesicular systems.

Liposomal commercial products are currently produced for different areas of use (liposomal drugs [135] liposomal cosmeceutical [136], etc.). For this reason, the industry is more familiar with liposome preparation techniques, precautions to be taken against possible problems, and quality control parameters.Both academic and industry knowledge of liposome technology facilitates technology transfer between units.Since there is less knowledge in other vesicular systems, it is more difficult to start industrial production and monitor the process. Although there are products of other vesicular systems in the cosmeceutical market [136], it takes a certain amount of time and effort to transfer this technology to the pharmaceutical industry.On the other hand, although the developed ocular drug delivery systems have achieved successful results in in vitro, ex vivo, and in vivo studies, obtaining ethics committee approvals for clinical studies is not an easy process. It is a long process for other vesicular systems to obtain the necessary permissions, complete the clinical trial processes, obtain regulatory approval, and be launched as a commercial product.

Although commercial vesicular ocular drug delivery system products are liposomal, the potential for the commercialization of other vesicular systems cannot be denied. Systems that are particularly applicable in the cosmetic industry are more likely to be producible in the pharmaceutical industry. Undoubtedly, the potential for the emergence of commercial products will lead to more scientific studies in this field. It is clear that vesicular systems will be used with increasing popularity in ocular drug targeting in the coming years. It is also possible to develop new vesicular systems with different structures than what is currently known with modifications.

## 7. Conclusions

Even though ocular diseases affect the quality of life of a large population worldwide, effective drug delivery cannot be achieved due to the anatomical structure of the eye. Topically applied dosage forms such as eye drops, ointments, and suspensions have a short residence time in the eye and therefore low bioavailability, while IVT, SCT, and suprachoroidal injections also cause various side effects with systemic use. Vesicular drug delivery systems, liposome, niosome, ethosome, transfersome and others (bilosome, cubosome, proniosome, chitosome, terpesome, phytosome, discome and spanlastics, flexosome, phytocubosome, and oleophytocubosome) provide alternative approach for efficient ocular drug delivery due to their advantages, including enhanced permeability, increased retention, improved solubility, reduced toxicity, targeted delivery, and higher bioavailability of the drug. Overall, this article has shown that the use of vesicular systems in ocular delivery has delivered considerable promise. These systems also provide prolonged effective drug release in the patient, thus increasing patient compliance compared to conventional drug delivery. As more commercial preparations become available, vesicular carriers are supposed to take the place of conventional systems.

## Figures and Tables

**Figure 1 pharmaceuticals-17-00511-f001:**
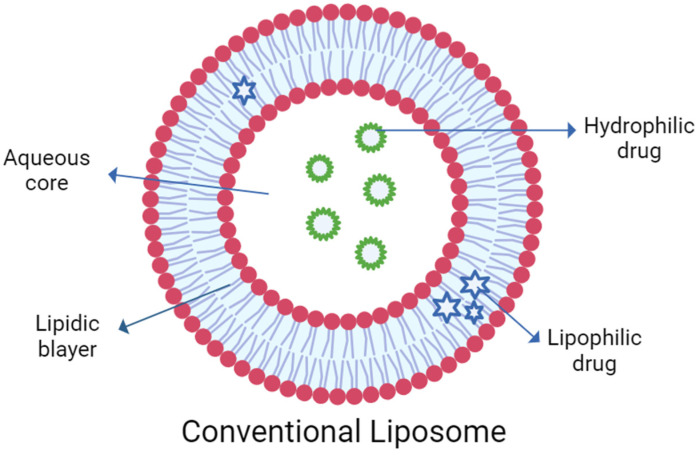
Entrapment of lipophilic drugs in vesicular delivery systems.

**Figure 2 pharmaceuticals-17-00511-f002:**
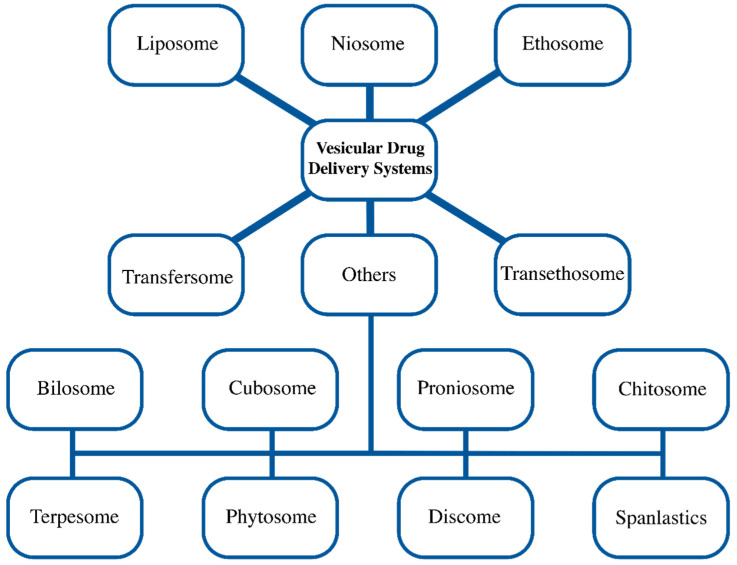
Different vesicular drug delivery systems.

**Table 1 pharmaceuticals-17-00511-t001:** Summary of active ingredients, therapeutic effect, lipids/surfactants, preparation method, and result of liposomes for ocular drug delivery over 5 years.

Active Ingredient	Therapeutic Effect	Lipids/Surfactants	Preparation Method	Result	Ref.
Rebamipide (RBM)	Dry eye	Hydrogenated soybean phospholipids (HSPC) and high purity cholesterol	Remote loading technique	Enhanced retention time at the corneal surface and improved drug penetration. Compared with suspension, Mucosta^®^, the longer retention time at the cornea allows liposomes to maintain a high concentration in the cornea and aqueous humor for a long time.	[47]
Ampicillin and ofloxacin	Stop ocular post-surgery infections	Soybean L-α PC from egg yolk (PC, 60% purity) while the rest was composed of a mixture of similar double-tailored phospholipids	Supercritical Assisted Liposome formation (SuperLip)	Ofloxacin and ampicillin exhibited encapsulation efficiencies of up to 97% and 99%, respectively. Compared to conventional techniques, a new selection of SuperLip process ensures higher encapsulation efficiency. High PC/H_2_O ratios produced higher EE thanks to longer water droplets flying time in the formation vessel and better lipid coverage.	[48]
Fluconazole (FLZ)	Fungal keratitis treatment	Lipoid S100 (PC from soybean)/Tween 80, Transcutol HP, and Caproyl 90	TFH	Enhanced corneal permeability. Ex vivo cumulative corneal permeation of FLZ after 6 h from HYS7, was 2.99 and 4.18 folds higher than conventional liposomes and FLZ suspension, respectively. In vivo corneal permeation of HYS7 showed sustained effect of FLZ reaching 24 h.	[49]
Ibuprofen	Corestenoma induced by cataract removal surgery	Purified soybean lecithin (PC S100, P94%, PC, approximately 70% linoleic acid, 8% lineolic acid, 5% oleic acid, 13% palmitic acid and 4% stearic acid residues) and stearylamine (SA)	Ethanol injection method	More sustained drug release behavior, and the release rates reduced as the SF concentration increased.	[8]
Timolol maleate (TML)	Glaucoma	Soybean phosphatidylcholine (SPC) and cholesterol	Ammonium sulfate gradient coupled with pH-gradient method	CH coating enabled sustained retention in the precornea, providing sustained action to increase drug permeability and bioavailability.	[9]
Bevacizumab	Ocular angiogenesis	1,2-dipalmitoyl-sn- glycero-3-phosphoethanolamine-N-[methoxy(polyethylene glycol)- 2000] (ammonium salt) (DPPE-PEG2000) and 1,2-dipalmitoyl-sn-gly- cero-3-phospho-(1′-rac-glycerol) (sodium salt) (DPPG), DOPE, cholesterol/PEG200	Modified lipid hydration and extrusion methods	Slow release and sustained anti-VEGF activity.	[50]
Distamycin A (DA)	Acyclovir-resistant HSV keratitis	Phosphatidylcholine (PC), Cholesterol	REV	Enhanced the bioavailability. The in vivo investigations showed the high bioavailability of DA in tear fluid that at the same time allowed an appreciable uptake of drug into the cornea up to concentration values able to produce the inhibition of viral replication (IC50) and without any evidence of transcorneal permeation.	[51]
siRNA	Acanthamoeba keratitis	1,2-di-(9E-octadecenoyl)-sn-glycero-3- phosphoethanolamine (DOPE) (Lipoid, GER), 1,2-dioleoylsn-glycero-3- trimethylammonium propane (DOTAP) (Lipoid, GER), and 1,2-distearoyl-sn-glycero-3-phosphoethanolamine-N-[methoxy(polyethylene glycol)-2000] (ammonium salt) (DSPE-PEG) (Lipoid, GER)	TFH	60% complete regression in corneal damage, without lymphocytic infiltrate.	[52]
Resveratrol	Blue-light-induced retinal damage	Cholesterol, Egg yolk phospholipid (EYPC)	Ethanol injection method	Trimethylated chitosan (TMC-coated) liposomes more easily penetrated the fundus than the bare flexible liposomes and aided in the enrichment of TMC-Lipo in the retina.	[53]
Sunitinib and Acriflavine	Choroidal neovascularization (CNV)	Lecithin, cholesterol/DSEP-PEG2000 and DSEP-PEG2000-cRGD	Ethanol injection method	Longer retention time in the target area, significant anti-CNV effect.	[54]
Triamcinolone acetonide	Macular edema	Soybean (PC), Coumarin6(C6) and cholesterol	Calcium acetate gradient method	High entrapment efficiency, exhibited a sustained release profile, and showed excellent physical stability.	[55]
Astragaloside IV (AS-IV) and tetramethylpyrazine (TMP)	AMD	Egg yolk lecithin and cholesterol/Poloxamers (P407 and P188), Propylene glycol, Gelucire44/14 and mPEG-CS	Ethanol injection method	Enhance the ocular bioavailability of AS-IV and TMP, which is the enhanced synergism of well-permeable liposome and slow-releasing hydrogel.	[56]
Travoprost (TRAVO)	Glaucoma and ocular hypertension	Soya bean lecithin, Cholesterol/Gellan gum and carbopol 934	TFH	Non-irritant, higher concentration of TRAVO in aqueous humor.	[57]
Ganciclovir (GCV)	Cytomegalovirus (CMV) retinitis	Cholesterol, 1,2-distearoyl-sn-glycero-3-phosphocholine (DSPC), DSPE-PEG, 1,2-distearoyl-sn-glycero-3-phosphoethanolamine-N-[maleimide (polyethylene glycol)-2000] (DSPEPEG-Mal)	REV	In vitro cytotoxicity test showed that formulations were safe for the ARPE-19 cells with percentage cell viability of 80–100% and they could inhibit the expression of CMV glycoprotein B after infection effectively.	[58]
Methazolamide (MTA)	Glaucoma	PC, Cholesterol	TFH	Longer precorneal residence time and ability to withstand drug release, better patient acceptance.	[59]
Fluconazole (FLZ)	Fungal keratitis	Phospholipon 90G (P-90G) and Cholesterol	TFH	Increased residence time, higher ex vivo permeation, no hemolysis, and ocular irritation was observed in a preclinical study.	[60]
Bevacizumab	Ocular angiogenesis	DPPC (1,2-dipalimitoyl-Sn-glycero-3-phosphocholine) and cholesterol	TFH	Improved the therapeutic application, and patient compliance thanks to small size, the ability of penetration through the cellular barrier, and its safety. The results showed the stability of BVZ after encapsulation in the nanoliposome	[61]
Moxifloxacin	Bacterial keratitis	Soy lecithin	Probe sonication	Improved particle size and homogeneity, 3D printed ocular inserts have high content uniformity and stability and controlled release.	[62]

**Table 2 pharmaceuticals-17-00511-t002:** Summary of active ingredient, preparation method, result, and therapeutic effect for vesicular systems (bilosome, cubosome, proniosome, chitosome, terpesome, phytosome, discome, spanlastics, flexosome, phytocubosome, and oleophytocubosome) for ocular drug delivery.

Delivery Systems	Active Ingredient	Therapeutic Effect	Lipids/Surfactants	Preparation Method	Result	Ref.
Bilosome	Acetazolamide (ACZ)	Glaucoma	Cholesterol/Span 60	TFH	Improved the bioavailability, reduced systemic absorption, and decreased the necessity for frequent administration, leading to enhanced patient compliance.	[87]
	Natamycin (NT)	Fungal keratitis and other fungal infections	Cholesterol/Span 60	TFH	Improved tear/corneal surface contact time and corneal permeability.	[88]
	Terconazole	Ocular fungal infections	Cholesterol/Span 60	Ethanol injection method	Ultradeformable bilosomes (UBs) contain an edge activator that imparts extra elasticity to the vesicles and consequently hypothesized to result in improved corneal permeation.	[89]
Cubosome	Voriconazole	Ocular fungal infections	DL-ά-Monoolein (MO), Pluronic F127 (F127)	Melt dispersion emulsification method	High mucoadhesive properties and enhanced precorneal residence time.	[90]
	Acetazolamide (ACZ)	Glaucoma	Glyceryl monooleate)(GMO)/Poloxamer 407) (P407)	Emulsification technique	Increased corneal permeability of ACZ.	[91]
	Latanoprost	Glaucoma	Phytantriol (3,7,11,15-tetramethyl-1,2,3-hexadecanetriol)/Pluronic F127	Bottom-up (BU) and top-down (TD) method	Demonstrated slow and sustained in vitro releasing profile of latanoprost.	[92]
Proniosome	Voriconazole	Fungal keratitis	Cholesterol, Span 60 (sorbitan monostearate), Span 80 (sorbitan monooleate) and pluronic F 127 (polox- amer 407)	Coacervation-phase method	Reduce the frequency of dosing intervals and improve patient compliance.	[93]
	Lomefloxacin HCl	Bacterial conjunctivitis	Cholesterol/Brij 35 (Polyoxyethylene (23) lauryl ether), Brij 72, Brij 98 Span 20, Span 40 Span 60, Tween 40, Tween 60, Tween 80	Coacervation-phase method	Enhances the retention of the drug maintaining a high local effect in the cornea, control the drug. release and the expected high stability on storage	[94]
	Dorzolamide HCI	Glaucoma	L-a-lecithin from soya bean, Span 40, cholesterol	Coacervation-phase method	Higher reduction in IOP, significantly sustaining and increasing Dorz bioavailability compared to Trusopt^®^ eye drops.	[95]
	Curcumin	Ocular inflammation	Span 60, and Cholesterol, Lecithin, Tween 80	Coacervation-phase method	Curcumin is a natural biofreindly alternative to an anti-inflammatory drug with fewer side effects. Selected proniosomal gel showed enhanced permeability 3.22-fold and 1.76-fold higher than curcumin dispersion and its lyophilized form, respectively.	[96]
Chitosome	Carteolol	Glaucoma	Cholesterol, span 60	TFH	Chitosan-coated carteolol niosome exhibited sustained in vitro drug release and enhanced chitosan permeation than chitosan solution.	[97]
	Ciprofloxacin	Bacterial conjunctivitis	Cholesterol, span 60	TFH	Enhanced the ocular retention time. The permeation study showed 1.79-fold enhancements in corneal permeation compared with marketed ciprofloxacin eye drop. HET-CAM study showed 0 scores (no irritation).	[98]
Terpesome	Fenticonazole nitrate	Ocular fungal infection	L-α phosphatidylcho- line (from egg yolk source)	TFH	High ocular retention, the in vivo study showed higher ocular retention of the optimized fenticonazole nitrate-loaded terpesomes relative to the drug suspension.	[99]
	Moxifloxacin hydrochloride	Bacterial keratitis	L-α phosphatidylcho- line (from egg yolk source)	TFH	Enhanced ocular drug conveyance	[100]
Phytosome	L-carnosine	Relief of dry eye conditions and for promoting healing after cataract and other refractive surgeries	Lipoid s 75	Solvent evaporation method	Enhanced corneal permeation	[101]
Discome	Naltrexone hydrochloride (NTX)	Diabetic keratopathy	Span 60, cholesterol	Reverse-phase evaporation (REV) method	Enhanced corneal uptake of the hydrophilic drug (NTX) and protected the encapsulated NTX from photo-oxidation compared with conventional NTX aqueous solutions.	[102]
Spanlastics	Clotrimazole	Ocular fungal infection	Tween 80, Kolliphor RH40 and Pluronic F127	Ethanol injection method	Optimum corneal permeability and elasticity.	[103]
	Miconazole nitrate	Ocular fungal infection	Cholesterol, tween 80, span 60	Ethanol injection method	Enhances the ocular permeability and bioavailability	[104]
Flexosome	Tolnaftate (TOL)	Ocular fungal infection	L-a-phosphatidylcholine from egg yolk and Tween 80	Ethanol injection method	Showed high encapsulation efficiency, small particle size, and spherical morphology, enhanced corneal permeation and antifungal activity.	[105]
Phytocubosome	Luteolin (LU)	Glaucoma and ocular inflammation	Glyceryl monooleate (GMO), Poloxamer 407, Phospholipid S100 (PL)	Hydrotrope technique	CH-coated phytocubosomes possessed improved transcorneal permeation, stronger anti-glaucomal action than uncoated phytocubosome, cubosome and suspension.	[106]
Oleophytocubosome	Luteolin (LU)	Glaucoma and ocular inflammation	Glyceryl monooleate (GMO), Poloxamer 407, Phospholipid S100 (PL)	Hydrotrope technique	Carrageenan-based ion-sensitive in situ gel (ISG) loaded with oleophytocubosome increases luteolin solubility and ocular bioavailability	[107]

**Table 3 pharmaceuticals-17-00511-t003:** Ocular commercial products containing the vesicular system.

Vesicular System	Commercial Name	Dosage Form	Disease	Company	References
Liposome	NaviLipo^®^	Drop	Dry eye syndrome	Novax Pharma	[127]
	VisuEvo^®^			Visufarma	[128]
	Eye Logic	Spray		Savant Health	[129]
	Perfect Liposomal			MPG&E	[130]
	Tears Again			BioRevive	[131]
	Occuvers Hyaluron			Innomedis	[132]
	Occuvers Lipostamin				[133]
	Retaine Liposome			OcuSoft	[134]

## Data Availability

Data sharing is not applicable.

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
