# Peer review of "Vesicular Drug Delivery Systems: Promising Approaches in Ocular Drug Delivery"

_pharmaceuticals, 2024, doi:10.3390/ph17040511_

Round 1

Reviewer 1 Report

Comments and Suggestions for Authors

While the review draft provides a constructive and comprehensive evaluation of the manuscript "Vesicular Drug Delivery Systems and Their Use in the Treatment of Ocular Diseases," there are areas where the review could be improved for clarity, objectivity, and depth. Below are some critiques and suggestions for enhancing the quality of the peer review:

1. This review mentions the advantages and potential of various vesicular systems but does not provide specific examples or references from the manuscript to support these claims. This lack of evidence makes it difficult to gauge the accuracy and depth of the manuscript's analysis.

2. The review often makes broad statements about the benefits of vesicular drug delivery systems without critically analyzing the limitations, potential risks, or the variability in efficacy among different vesicular formulations. Therefore, Incorporate a more balanced perspective by discussing potential drawbacks or limitations of vesicular systems as presented in the manuscript. Highlight any gaps in research, discrepancies in data, or areas where the conclusions drawn by the authors may not be fully supported by the evidence provided.

3. Is there any marketed products from these systems? If yes, need to list down in a table form. IF there is no product yet, then need to explain what are the barriers to bring the product into the market.

4. The reviewer should critically evaluate the manuscript's contribution to existing knowledge (should be highlighted in introduction). If similar reviews exist in the literature, the reviewer should assess how this manuscript adds value or presents a unique perspective on the topic.

5. The review briefly mentions the systematic approach employed in the manuscript but does not assess the rigor of the research methodology, such as the criteria for selecting studies for inclusion, the assessment of study quality, or the synthesis of findings

6. Additional figures comparing the structural differences, some in vivo study figures and tables summarizing clinical studies on the vesicular systems reviewed could enhance the manuscript's readability and impact.

7. Limitation of nanoparticle systems for ouclar application should be discussed, considering some of these points: The limitations of nanoparticles in ocular applications include potential toxicity to ocular tissues, challenges in penetrating the eye's protective barriers to reach the target site effectively, and difficulties in controlling the release and distribution of the therapeutic agent. Additionally, the long-term stability and the risk of aggregation of nanoparticles can further complicate their use in eye-related treatments.

I recommend the manuscript for publication after major revisions are made to address the points mentioned above.

Comments on the Quality of English Language

Some sentences are complex and could be simplified for readability. A thorough proofreading can help catch and correct some of minor writing issues.

Author Response

Dear Reviewer,

On behalf of my colleagues and myself, I would like to extend our sincerest appreciation for your invaluable contributions. Your insightful feedback has been instrumental in refining our manuscript. We have endeavored to provide the best possible responses to the requests you made in order to enhance our manuscript. We trust that the modifications we have implemented will resonate meaningfully with you. On behalf of all my colleagues, I extend our appreciation for considering the modifications we have implemented.

Respectfully submitted.

1. This review mentions the advantages and potential of various vesicular systems but does not provide specific examples or references from the manuscript to support these claims. This lack of evidence makes it difficult to gauge the accuracy and depth of the manuscript's analysis.

The authors thank the reviewer for his/her comments and contributions. The purpose of writing this review is to inform the reader about the ocular use of vesicular systems. For this purpose, we first describe the various vesicular systems we have mentioned. For a better understanding by the reader, we have summarized the articles studied in the field in a table and cited them at the end of each line to support these claims. Taking into account the reviewer's suggestion, specific examples are given in the table of the results of the given articles on the ocular use of vesicular systems. References and added examples are highlighted in the table.

2. The review often makes broad statements about the benefits of vesicular drug delivery systems without critically analyzing the limitations, potential risks, or the variability in efficacy among different vesicular formulations. Therefore, Incorporate a more balanced perspective by discussing potential drawbacks or limitations of vesicular systems as presented in the manuscript. Highlight any gaps in research, discrepancies in data, or areas where the conclusions drawn by the authors may not be fully supported by the evidence provided.

The authors thank the reviewer for his/her comments and contributions. Taking into account the reviewer's suggestion, the disadvantages and limitations associated with the use of vesicular systems are mentioned in the article, and references and datas can be found in the review in highlighted.

The ocular use of vesicular systems has many advantages as we have mentioned. In addition to all these advantages, there are disadvantages such as oxidation and hydrolysis-like reaction, low solubility, leakage and fusion of encapsulated drug/molecules, stability, high production cost, short half-life and sterilization problems [41]. Intraocular clouding may occur with administration via intravitreal injection. Liposomal formulations generally require more complicated clinical trials and an expensive manufacturing process than conventional formulations. In addition to general difficulties including short shelf life, low drug loading, and steri-lization issues, liposomes face ocular-specific difficulties such limited retention time for intravitreally injected liposomes and unclear ocular safety during prolonged and repeated use. These obstacles need to be cleared before liposomes are applied topically [42].For niosomes, insufficient clinical data available for surfactant related in vivo toxicity and irritation. Another vesicular systems, cubosomal formulations, are challenging to scale up and manufacture on large-scales because of their complicated phase behavior and viscosity-related complexity [3].

41. Akbarzadeh A, Rezaei-Sadabady R, Davaran S, Joo SW, Zarghami N, Hanifehpour Y, et al. Liposome: Classification, preparation, and applications. Nanoscale Res Lett. 2013;8(1). https://doi.org/10.1186%2F1556-276X-8-102.

42. Li Q, Weng J, Wong SN, Thomas Lee WY, Chow SF. Nanoparticulate Drug Delivery to the Retina. Vol. 18, Molecular Pharmaceutics. American Chemical Society; 2021. p. 506–21.

3.  Das B, Nayak AK, Mallick S. Lipid-based nanocarriers for ocular drug delivery: An updated review. Vol. 76, Journal of Drug Delivery Science and Technology. Editions de Sante; 2022. https://doi.org/10.1016/j.jddst.2022.103780.

3. Is there any marketed products from these systems? If yes, need to list down in a table form. IF there is no product yet, then need to explain what are the barriers to bring the product into the market.

The authors thank the reviewer for his/her comments and contributions. The products on the market are given in a table form under the title "Future Perspectives on Scientific and Commercial’’.

4. The reviewer should critically evaluate the manuscript's contribution to existing knowledge (should be highlighted in introduction). If similar reviews exist in the literature, the reviewer should assess how this manuscript adds value or presents a unique perspective on the topic.

The literature contribution of the article has been added under the title "introduction" and the changes have been colored.

There are articles in the literature on the ocular use of vesicular systems [6-8]. Most of these articles have been studied with liposomes [9,10]. However, in our study, different kinds of vesicular systems (liposome, niosome, ethosome, transfersome, and others (bilosome, transethosome, cubosome, proniosome, chitosome, terpesome, phytosome, discome, spanlastics, flexosomes, phytocubosome and oleophytocubosome) were comprehensively evaluated, including studies with different therapeutic agents, their therapeutic effects, production methods and formulation content (lipids/surfactans). 

6. Kaur IP, Garg A, Singla AK, Aggarwal D. Vesicular systems in ocular drug delivery: An overview. Vol. 269, International Journal of Pharmaceutics. Elsevier; 2004. p. 1–14.

7. Das B, Nayak AK, Mallick S. Lipid-based nanocarriers for ocular drug delivery: An updated review. J Drug Deliv Sci Technol [Internet]. 2022 Oct 1;76:103780. Available from: https://linkinghub.elsevier.com/retrieve/pii/S1773224722006918

8. Mosallam S, Albash R, Abdelbari MA. Advanced Vesicular Systems for Antifungal Drug Delivery. Vol. 23, AAPS PharmSciTech. Springer Science and Business Media Deutschland GmbH; 2022.

9. Dong Y, Dong P, Huang D, Mei L, Xia Y, Wang Z, et al. Fabrication and characterization of silk fibroin-coated liposomes for ocular drug delivery. European Journal of Pharmaceutics and Biopharmaceutics. 2015;91:82–90.

10. Tan G, Yu S, Pan H, Li J, Liu D, Yuan K, et al. Bioadhesive chitosan-loaded liposomes: A more efficient and higher permeable ocular delivery platform for timolol maleate. Int J Biol Macromol. 2017 Jan 1;94:355–63.

5. The review briefly mentions the systematic approach employed in the manuscript but does not assess the rigor of the research methodology, such as the criteria for selecting studies for inclusion, the assessment of study quality, or the synthesis of findings.

As we mentioned in our article introduction, this review aims to inform the reader and demonstrate the advantages of using vesicular systems in ocular drug delivery. The eye has always been an interesting and challenging field for pharmaceutical technologists due to its unique anatomical and physiological structure and limited bioavailability of conventional drugs. In establishing our research methodology, we first informed the reader about ocular anatomy and drug administration routes. We then provided general information about vesicular systems and summarized the studies in the field in tables with references. There are many articles in the literature on ocular drug delivery systems (1424 articles appear only in the last 5 years). Studies on vesicular systems, especially liposomes, are included in a large number of these articles. For this reason, we have filtered liposome-related studies in the last 5 years to include the current literature in our article. Other systems (niosome, ethosome, transfersome, and others (bilosome, transethosome, cubosome, proniosome, chitosome, terpesome, phytosome, discome, spanlastics, flexosomes, phytocubosome and oleophytocubosome) have been studied especially in the last 10 years.

6. Additional figures comparing the structural differences, some in vivo study figures and tables summarizing clinical studies on the vesicular systems reviewed could enhance the manuscript's readability and impact.

The authors thank the reviewer for his/her comments and contributions. Since figures and tables from other articles would be copyrighted, we have created the tables and drawn the figures ourselves in the article. Our aim is not to describe the work done in the field, but to summarize and provide general information to the reader. 

7. Limitation of nanoparticle systems for ocular application should be discussed, considering some of these points: The limitations of nanoparticles in ocular applications include potential toxicity to ocular tissues, challenges in penetrating the eye's protective barriers to reach the target site effectively, and difficulties in controlling the release and distribution of the therapeutic agent. Additionally, the long-term stability and the risk of aggregation of nanoparticles can further complicate their use in eye-related treatments.

The authors thank the reviewer for his/her comments and contributions. Taking into account the reviewer's suggestion, the disadvantages and limitations of nanoparticle systems are mentioned in the article, and references and datas can be found in the review in highlighted.

Nanoparticulate drug delivery system provide the capability for delivering therapeutics to the specific ocular targets. The use of polymeric systems (nano-emulsion, dendrimer, polymeric micelle, solid lipid nanoparticle, polymeric nanoparticle, micro- and nano-spheres) was investigated for ocular drug delivery. Although current research shows that nanoparticulate delivery methods have immense therapeutic potential, transferring these systems from bench to bedside represents a difficult challenge.  For dendrimers, only a few clinical trials have been launched, and no safety or tolerability outcome has been announced. Particle growth, unpredictable gelation tendency and unexpected polymorphic transition dynamics are some stability issues of solid lipid nanoparticles and polymeric nanoparticle. Long-term and repeated use of nanoparticle systems requires further research [42]. Among these carriers, liposomes have been most studied. Liposome includes an aqueous core entrapped by one or more bilayers composed of natural or synthetic lipids, which makes them biodegradable and biocompatible. They are composed of natural phospholipids that are biologically inert and they have low toxicity. Drugs with different lipophilicities can be encapsulated into liposomes: strongly lipophilic drugs are entrapped in the lipid bilayer, hydrophilic drugs are located in the aqueous core [43].

42. Li Q, Weng J, Wong SN, Thomas Lee WY, Chow SF. Nanoparticulate Drug Delivery to the Retina. Vol. 18, Molecular Pharmaceutics. American Chemical Society; 2021. p. 506–21.

43. Immordino ML, Dosio F, Cattel L. Stealth liposomes: review of the basic science, rationale, and clinical applications, existing and potential. Int J Nanomedicine. 2006;1(3):297–315.

Best regards.

Reviewer 2 Report

Comments and Suggestions for Authors

Review of article. “Vesicular Drug Delivery Systems and Their Use in the Treatment of Ocular Diseases

1.      As the review article reports current trends in vesicular drug delivery systems for ocular route, the title should also reflect the same. The article does not elaborate on challenges and formulation approaches for various ocular diseases and mainly emphasizes on ocular drug delivery systems, indicating mention of “treatment of ocular diseases” in title irrelevant. Title can be modified to make it more suitable to the scope of this article.

2.      Abstract is comprehensive and summarises the objective of review. Few minor changes are advisable. Line 28 in abstract- Instead of word, ‘general information’, ‘their applications in management of ocular diseases’ or similar other approapriate term can be added. 

3.      The initial sections of the review article are well written and informative, including major ocular barriers for drug permeation. However, as the title of the review indicates focus on ocular diseases, any physiological changes pertaining to various ocular diseases and their effect on drug delivery and permeation is not included in the review.

4.      The sections related to vesicular carriers are generalised. The section only describes the work done by different researchers on various vesicular carriers. There is no specific information or separate discussion on different methodologies, lipids or formulation or processing variables affecting the drug delivery system which in turn affects their efficacy.   

5.      Various formulations approaches used to modify the surface of vesicular carriers for targeting of actives to various ocular regions also need to be discussed.

6.      A section on future perspective is advisable.

Overall, the review article is interesting and will be useful for researchers working in the area.  The article can be accepted upon revision and addressing the above points.

Comments on the Quality of English Language

The quality of English in the manuscript is acceptable. Authors should address few minor grammatical errors in the manuscript.

Author Response

Dear Reviewer,

On behalf of myself and my colleagues, I would like to extend our sincere appreciation for the valuable contributions you have made. We have diligently worked to provide the most comprehensive responses to your requests in order to enhance our manuscript. We trust that the revisions we have made will be meaningful from your perspective as well. Please accept our gratitude for considering the adjustments we have incorporated.

Respectfully submitted for your consideration.

1. As the review article reports current trends in vesicular drug delivery systems for ocular route, the title should also reflect the same. The article does not elaborate on challenges and formulation approaches for various ocular diseases and mainly emphasizes on ocular drug delivery systems, indicating mention of “treatment of ocular diseases” in title irrelevant. Title can be modified to make it more suitable to the scope of this article.

The authors thank the reviewer for his/her comments and contributions. The title has been changed upon your suggestion. The new title is: ‘’Vesicular Drug Delivery Systems: Promising Approaches in Ocular Drug Delivery’’.

2. Abstract is comprehensive and summarises the objective of review. Few minor changes are advisable. Line 28 in abstract- Instead of word, ‘general information’, ‘their applications in management of ocular diseases’ or similar other approapriate term can be added.

The word was changed on your suggestion instead of ‘’generel information’’ to ‘’their applications in management of ocular diseases’’.

3. The initial sections of the review article are well written and informative, including major ocular barriers for drug permeation. However, as the title of the review indicates focus on ocular diseases, any physiological changes pertaining to various ocular diseases and their effect on drug delivery and permeation is not included in the review.

The title has been changed upon your suggestion 1. The new title is: ‘’Vesicular Drug Delivery Systems: Promising Approaches in Ocular Drug Delivery’’.

4. The sections related to vesicular carriers are generalised. The section only describes the work done by different researchers on various vesicular carriers. There is no specific information or separate discussion on different methodologies, lipids or formulation or processing variables affecting the drug delivery system which in turn affects their efficacy.

The authors thank the reviewer for his/her comments and contributions. Taking into account the reviewer's suggestion,  the information we have added to our article is under the title ‘’4.1 Liposome’’, ‘’4.2 Niosome’’, ‘’4.3 Ethosome/Transethosome’’ and the changes have been colored. We have given specific information and discussion for liposomes, niosomes and ethosomes/transethosomes, since there is no fixed parameter to compare, for other vesicular systems that have different components and transfersomes do not have a different production method.

5. Various formulations approaches used to modify the surface of vesicular carriers for targeting of actives to various ocular regions also need to be discussed.

The authors thank the reviewer for his/her comments and contributions. Taking into account the reviewer's suggestion,  the information we have added to our article is under the title "5. Functionalization of Vesicular Systems’’ and the changes have been colored.

6. A section on future perspective is advisable.

The authors thank the reviewer for his/her comments and contributions. Taking into account the reviewer's suggestion,  future perspective has added to our article.

Sincerely yours,

Reviewer 3 Report

Comments and Suggestions for Authors

The authors present a review of liposomes and other somes-based drug delivery systems. The idea of the article is good, but they really need to state at the beginning, that the authors only focus on a review of the last 5 years. Furthermore, there are a lot of unproven claims that reduce the quality of this manuscript at the moment. The authors need to make clear which sentence and claim refers to which reference. For this reason, and the number of claims, I suggest a major revision to give the authors enough time to improve their manuscript.

1.      Page 1 line 37 reference missing.

2.      Page 1 line 42 reference missing.

3.      Page 1 line 43 reference missing.

4.      Page 2 line 46 sentence 1 reference missing.

5.      Page 2 line 48 reference missing.

6.      Page 2 line 51 reference missing.

7.      Page 2 line 54 reference missing.

8.      Page 2 line 63 reference missing.

9.      Page 2 line 64 reference missing.

10.   Page 2 line 65 reference missing.

11.   Page 2 line 67 reference missing.

12.   Page 2 line 72 reference missing.

13.   Page 2 line 73 reference missing.

14.   Page 2 line 74 reference missing.

15.   Page 2 line 77 reference missing.

16.   Page 2 line 78 reference missing.

17.   Page 2 line 79 reference missing.

18.   Page 2 line 80 reference missing.

19.   Page 2 line 83 reference missing.

20.   Page 2 line 85 reference missing.

21.   Page 2 line 89 reference missing.

22.   Page 2 line 93 reference missing.

23.   Page 2 line 96 reference missing.

24.   Page 2 line 97 reference missing.

25.   Page 2 line 98 reference missing.

26.   Page 3 line 99 reference missing.

27.   Page 3 line 101 reference missing.

28.   Page 3 line 102 reference missing.

29.   Page 3 line 106 reference missing.

30.   Page 3 line 108 reference missing.

31.   Page 3 line 110 reference missing.

32.   Page 3 line 114 reference missing.

33.   Page 3 line 119 reference missing.

34.   Page 3 line 120 reference missing.

35.   Page 3 line 122 reference missing.

36.   Page 3 line 124 reference missing.

37.   Page 3 line 128 reference missing.

38.   Page 3 line 129 reference missing.

39.   Page 3 line 131 reference missing.

40.   Page 3 line 132 reference missing.

41.   Page 3 line 134 reference missing, where is the evidence?

42.   Page 3 line 139 reference missing.

43.   Page 3 line 143 reference missing.

44.   Page 3 line 145 reference missing.

45.   Page 4 line 154 reference missing.

46.   Page 4 line 156 reference missing.

47.   Page 4 line 160 reference missing.

48.   Page 4 line 161 reference missing.

49.   Page 4 line 166 reference missing.

50.   Page 4 line 168 reference missing.

51.   Page 4 line 171 reference missing.

52.   Page 4 line 174 reference missing.

53.   Page 4 line 176 reference missing.

54.   Page 4 line 180 reference missing.

55.   Page 4 line 181 reference missing.

56.   Page 4 line 183 reference missing.

57.   Page 4 line 184 reference missing.

58.   Page 4 line 185 reference missing.

59.   Page 4 line 190 reference missing.

60.   Page 4 line 198 reference missing.

61.   Page 4 line 202 reference missing.

62.   Page 4 line 204 reference missing.

63.   Page 4 line 205 reference missing.

64.   Page 5 line 206 reference missing.

65.   Page 5 line 211 reference missing.

66.   Page 5 line 213 reference missing.

67.   Page 5 line 214 reference missing.

68.   Page 5 line 216 reference missing.

69.   Page 5 line 219 reference missing.

70.   Page 5 line 223 reference missing.

71.   Page 5 line 225 reference missing.

72.   Page 5 line 228 reference missing.

73.   Page 5 line 230 reference missing.

74.   Page 5 line 231 reference missing.

75.   Page 5 line 236 reference missing.

76.   Page 5 line 240 reference missing.

77.   Page 5 line 243 reference missing.

78.   Page 5 line 252 reference missing.

79.   Page 5 line 254 reference missing.

80.   Page 5 line 256 reference missing.

81.   Page 6 line 269 reference missing

82.   Page 6 line 273 reference missing, here following polymer system reference might be useful: 1

83.   Modern polymer drug delivery systems can be produced fully automated and are a promising outlook for application in this field.2

84.   Page 7 line 298 reference missing

85.   Page 7 line 301 reference missing

86.   Page 7 line 313 reference missing

87.   Page 7 line 314 references for the two sentences missing

88.   Page 7 line 319 reference missing

89.   Page 7 line 320 reference missing

90.   Page 7 line 321 reference missing

91.   Page 8 line 327 reference missing

92.   Page 8 line 340 2 of CO2 is not subscript

93.   Page 12 line 439 reference missing.

94.   Page 12 line 440 reference missing.

95.   Page 12 line 441 reference missing.

96.   Page 12 line 443 reference missing.

97.   Page 12 line 446 reference missing.

98.   Page 12 line 440 2 of CO2 not subscript.

99.   English grammar must be improved in some cases.

References

(1)        Sukhorukov, G. B.; Donath, E.; Lichtenfeld, H.; Knippel, E.; Knippel, M.; Budde, A.; Mohwald, H. Layer-by-Layer Self Assembly of Polyelectrolytes on Colloidal Particles. Colloids Surfaces A Physicochem. Eng. Asp. 1998, 137, 253–266. https://doi.org/10.1016/S0927-7757(98)00213-1.

(2)        Li, W.; Gai, M.; Rutkowski, S.; He, W.; Meng, S.; Gorin, D.; Dai, L.; He, Q.; Frueh, J. An Automated Device for Layer-by-Layer Coating of Dispersed Superparamagnetic Nanoparticle Templates. Colloid J. 2018, 80 (6), 648–659. https://doi.org/10.1134/S1061933X18060078.

Comments on the Quality of English Language

1.      English grammar must be improved in some cases.

Author Response

Dear Reviewer,

We wish to express our sincere appreciation for your valuable contributions to our manuscript. Your insights and suggestions have played a crucial role in refining our work. We have diligently addressed the requests you made, endeavoring to enhance the quality and clarity of our manuscript to the best of our abilities. We trust that the revisions we've made meet your expectations and contribute to the overall improvement of the manuscript. On behalf of all the authors, we extend our heartfelt gratitude for your time and expertise in reviewing our work.

Respectfully submitted for your consideration.

The authors present a review of liposomes and other somes-based drug delivery systems. The idea of the article is good, but they really need to state at the beginning, that the authors only focus on a review of the last 5 years. Furthermore, there are a lot of unproven claims that reduce the quality of this manuscript at the moment. The authors need to make clear which sentence and claim refers to which reference. For this reason, and the number of claims, I suggest a major revision to give the authors enough time to improve their manuscript.

-The authors thank the reviewer for his/her comments and contributions. We have done our article filtering for liposome in the last 5 years. Taking into account the reviewer's suggestion, we changed the sentence as ‘’For this reason, only studies from the last 5 years have been evaluated for liposome in this review’’. Since the articles discussed in the last 5 years are related to liposomes, this information is given under the title "Liposome".

-In fact, all the information we provide is referenced and there are nounproven claims. According to article citation rules, multiple sentences or information taken from the same article may be cited at the end of these sentences or at the end of the first sentence mentioned. There is no need to cite again at the end of each sentence.

-Addition referances have been made to the parts you have warned, but we think there is a shift in some lines on your word format and ours. You mentioned that ‘’Request 81: Page 6 line 269 reference missing’’, but we see that there is a ‘’Vesicular System’’ title in the same line.

-Request 41: We changed the sentence as ‘’ Knowledge of the literature indicates that the ciliary body and iris express specific active drug transporters, thereby hindering the permeation of drugs.’’

- Request 82: In line 273, we mentioned that vesicular systems can encapsulate hydrophilic and hydrophobic substances thanks to their chemical structure (aqueous core and lipid bilayer) and lipid-embedded drugs can easily pass through membranes. The authors thank the reviewer for his/her contribution but polymer system reference irrelevant our subject in this line. The vesicular systems discussed in the article are not polymeric but lipid-based systems. The conjuctions link the requested citations at the end of the relevant sentence.

- Request 92: In line 340, CO2 is corrected as ‘’CO2’’.

- Request 98: In line 450, CO2 is corrected as ‘’CO2’’.

Best regards.

Round 2

Reviewer 3 Report

Comments and Suggestions for Authors

My comments from the first revision round were completly ignored by the reviewers. That is why again:

The authors present a review of liposomes and other somes-based drug delivery systems. The idea of the article is good, but they really need to state at the beginning, that the authors only focus on a review of the last 5 years. Furthermore, there are a lot of unproven claims that reduce the quality of this manuscript at the moment. The authors need to make clear which sentence and claim refers to which reference. For this reason, and the number of claims, I suggest a major revision to give the authors enough time to improve their manuscript.

1.       Page 1 line 37 reference missing.

2.       Page 1 line 42 reference missing.

3.       Page 1 line 43 reference missing.

4.       Page 2 line 46 sentence 1 reference missing.

5.       Page 2 line 48 reference missing.

6.       Page 2 line 51 reference missing.

7.       Page 2 line 54 reference missing.

8.       Page 2 line 63 reference missing.

9.       Page 2 line 64 reference missing.

10.   Page 2 line 65 reference missing.

11.   Page 2 line 67 reference missing.

12.   Page 2 line 72 reference missing.

13.   Page 2 line 73 reference missing.

14.   Page 2 line 74 reference missing.

15.   Page 2 line 77 reference missing.

16.   Page 2 line 78 reference missing.

17.   Page 2 line 79 reference missing.

18.   Page 2 line 80 reference missing.

19.   Page 2 line 83 reference missing.

20.   Page 2 line 85 reference missing.

21.   Page 2 line 89 reference missing.

22.   Page 2 line 93 reference missing.

23.   Page 2 line 96 reference missing.

24.   Page 2 line 97 reference missing.

25.   Page 2 line 98 reference missing.

26.   Page 3 line 99 reference missing.

27.   Page 3 line 101 reference missing.

28.   Page 3 line 102 reference missing.

29.   Page 3 line 106 reference missing.

30.   Page 3 line 108 reference missing.

31.   Page 3 line 110 reference missing.

32.   Page 3 line 114 reference missing.

33.   Page 3 line 119 reference missing.

34.   Page 3 line 120 reference missing.

35.   Page 3 line 122 reference missing.

36.   Page 3 line 124 reference missing.

37.   Page 3 line 128 reference missing.

38.   Page 3 line 129 reference missing.

39.   Page 3 line 131 reference missing.

40.   Page 3 line 132 reference missing.

41.   Page 3 line 134 reference missing, where is the evidence?

42.   Page 3 line 139 reference missing.

43.   Page 3 line 143 reference missing.

44.   Page 3 line 145 reference missing.

45.   Page 4 line 154 reference missing.

46.   Page 4 line 156 reference missing.

47.   Page 4 line 160 reference missing.

48.   Page 4 line 161 reference missing.

49.   Page 4 line 166 reference missing.

50.   Page 4 line 168 reference missing.

51.   Page 4 line 171 reference missing.

52.   Page 4 line 174 reference missing.

53.   Page 4 line 176 reference missing.

54.   Page 4 line 180 reference missing.

55.   Page 4 line 181 reference missing.

56.   Page 4 line 183 reference missing.

57.   Page 4 line 184 reference missing.

58.   Page 4 line 185 reference missing.

59.   Page 4 line 190 reference missing.

60.   Page 4 line 198 reference missing.

61.   Page 4 line 202 reference missing.

62.   Page 4 line 204 reference missing.

63.   Page 4 line 205 reference missing.

64.   Page 5 line 206 reference missing.

65.   Page 5 line 211 reference missing.

66.   Page 5 line 213 reference missing.

67.   Page 5 line 214 reference missing.

68.   Page 5 line 216 reference missing.

69.   Page 5 line 219 reference missing.

70.   Page 5 line 223 reference missing.

71.   Page 5 line 225 reference missing.

72.   Page 5 line 228 reference missing.

73.   Page 5 line 230 reference missing.

74.   Page 5 line 231 reference missing.

75.   Page 5 line 236 reference missing.

76.   Page 5 line 240 reference missing.

77.   Page 5 line 243 reference missing.

78.   Page 5 line 252 reference missing.

79.   Page 5 line 254 reference missing.

80.   Page 5 line 256 reference missing.

81.   Page 6 line 269 reference missing

82.   Page 6 line 273 reference missing, here following polymer system reference might be useful: 1

83.   Modern polymer drug delivery systems can be produced fully automated and are a promising outlook for application in this field.2

84.   Page 7 line 298 reference missing

85.   Page 7 line 301 reference missing

86.   Page 7 line 313 reference missing

87.   Page 7 line 314 references for the two sentences missing

88.   Page 7 line 319 reference missing

89.   Page 7 line 320 reference missing

90.   Page 7 line 321 reference missing

91.   Page 8 line 327 reference missing

92.   Page 8 line 340 2 of CO2 is not subscript

93.   Page 12 line 439 reference missing.

94.   Page 12 line 440 reference missing.

95.   Page 12 line 441 reference missing.

96.   Page 12 line 443 reference missing.

97.   Page 12 line 446 reference missing.

98.   Page 12 line 440 2 of CO2 not subscript.

99.   English grammar must be improved in some cases.

References

(1)         Sukhorukov, G. B.; Donath, E.; Lichtenfeld, H.; Knippel, E.; Knippel, M.; Budde, A.; Mohwald, H. Layer-by-Layer Self Assembly of Polyelectrolytes on Colloidal Particles. Colloids Surfaces A Physicochem. Eng. Asp. 1998, 137, 253–266. https://doi.org/10.1016/S0927-7757(98)00213-1.

(2)         Li, W.; Gai, M.; Rutkowski, S.; He, W.; Meng, S.; Gorin, D.; Dai, L.; He, Q.; Frueh, J. An Automated Device for Layer-by-Layer Coating of Dispersed Superparamagnetic Nanoparticle Templates. Colloid J. 2018, 80 (6), 648–659. https://doi.org/10.1134/S1061933X18060078.

Comments on the Quality of English Language

English grammar must be improved in some cases.

Author Response

Dear Reviewer,

The revisions as per your instructions from the previous revision cycle have been implemented (highlighted as yellow in the revised manuscript). 

The revised manuscript was resubmitted for your consideration.

Sincerely yours.

Answers:

-The authors thank the reviewer for his/her comments and contributions. We have done our article filtering for liposome in the last 5 years. Taking into account the reviewer's suggestion, we changed the sentence as ‘’For this reason, only studies from the last 5 years have been evaluated for liposome in this review’’. Since the articles discussed in the last 5 years are related to liposomes, this information is given under the title "Liposome".

-The lines that you have warned to add references are given below with the line numbers in the last updated version. Added references are highlighted in manuscript.

  1. Page 1 line 37 reference added.
  2. Page 1 line 42 reference added.
  3. Page 1 line 43 reference added. (line 44 in last version of manuscript)
  4. Page 2 line 46 sentence 1 reference added.
  5. Page 2 line 48 reference added. (line 50 in last version of manuscript)
  6. Page 2 line 51 reference added. (line 53 in last version of manuscript)
  7. Page 2 line 54 reference added. (line 55 in last version of manuscript)
  8. Page 2 line 63 reference added. (line 70 in last version of manusript)
  9. Page 2 line 64 reference added. (line 71 in last version of manusript)
  10. Page 2 line 65 reference added. (line 72 in last version of manusript)
  11. Page 2 line 67 reference added. (line 74 in last version of manusript)
  12. Page 2 line 72 reference added. (line 80 in last version of manusript)
  13. Page 2 line 73 reference added. (line 81 in last version of manusript)
  14. Page 2 line 74 reference added. (line 82 in last version of manusript)
  15. Page 2 line 77 reference added. (line 85 in last version of manusript)
  16. Page 2 line 78 reference added. (line 86 in last version of manusript)
  17. Page 2 line 79 reference added. (line 88 in last version of manusript)
  18. Page 2 line 80 reference added. (line 89 in last version of manusript)
  19. Page 2 line 83 reference added. (line 92 in last version of manusript)
  20. Page 2 line 85 reference added. (line 94 in last version of manusript)
  21. Page 2 line 89 reference added. (line 98 in last version of manusript)
  22. Page 2 line 93 reference added. (line 102 in last version of manusript
  23. Page 2 line 96 reference added. (line 106 in last version of manusript)
  24. Page 2 line 97 reference added. (line 107 in last version of manusript)
  25. Page 2 line 98 reference added. (line 108 in last version of manusript)
  26. Page 3 line 99 reference added. (line 110 in last version of manusript)
  27. Page 3 line 101 reference added. (line 112 in last version of manusript)
  28. Page 3 line 102 reference added. (line 114 in last version of manusript)
  29. Page 3 line 106 reference added. (line 115 in last version of manusript)
  30. Page 3 line 108 reference added. (line 117 in last version of manusript)
  31. Page 3 line 110 reference added. (line 119 in last version of manusript)
  32. Page 3 line 114 reference added. (line 123 in last version of manusript)
  33. Page 3 line 119 reference added. (line 128 in last version of manusript)
  34. Page 3 line 120 reference added. (line 129 in last version of manusript)
  35. Page 3 line 122 reference added. (line 131 in last version of manusript)
  36. Page 3 line 124 reference added. (line 133 in last version of manusript)
  37. Page 3 line 128 reference added. (line 137 in last version of manusript)
  38. Page 3 line 129 reference added. (line 138 in last version of manusript)
  39. Page 3 line 131 reference added. (line 140 in last version of manusript)
  40. Page 3 line 132 reference added. (line 141 in last version of manusript)
  41. Page 3 line 134 reference added. (line 143 in last version of manusript)
  42. Page 3 line 139 reference added. (line 149 in last version of manusript)
  43. Page 3 line 143 reference added. (line 153 in last version of manusript)
  44. Page 3 line 145 reference added. (line 156 in last version of manusript)
  45. Page 4 line 154 reference added. (line 165 in last version of manusript)
  46. Page 4 line 156 reference added. (line 167 in last version of manusript)
  47. Page 4 line 160 reference added. (line 171 in last version of manusript)
  48. Page 4 line 161 reference added. (line 172 in last version of manusript)
  49. Page 4 line 166 reference added. (line 178 in last version of manusript)
  50. Page 4 line 168 reference added. (line 180 in last version of manusript)
  51. Page 4 line 171 reference added. (line 183 in last version of manusript)
  52. Page 4 line 174 reference added. (line 184 in last version of manusript)
  53. Page 4 line 176 reference added. (line 188 in last version of manusript)
  54. Page 4 line 180 reference added. (line 192 in last version of manusript)
  55. Page 4 line 181 reference added. (line 193 in last version of manusript)
  56. Page 4 line 183 reference added. (line 195 in last version of manusript)
  57. Page 4 line 184 reference added. (line 196 in last version of manusript)
  58. Page 4 line 185 reference added. (line 197 in last version of manusript)
  59. Page 4 line 190 reference added. (line 202 in last version of manusript)
  60. Page 4 line 198 reference added. (line 211 in last version of manusript)
  61. Page 4 line 202 reference added. (line 216 in last version of manusript)
  62. Page 4 line 204 reference added. (line 217 in last version of manusript)
  63. Page 4 line 205 reference added. (line 218 in last version of manusript)
  64. Page 5 line 206 reference added. (line 219 in last version of manusript)
  65. Page 5 line 211 reference added. (line 224 in last version of manusript)
  66. Page 5 line 213 reference added. (line 226 in last version of manusript)
  67. Page 5 line 214 reference added. (line 228 in last version of manusript)
  68. Page 5 line 216 reference added. (line 230 in last version of manusript)
  69. Page 5 line 219 reference added. (line 232 in last version of manusript)
  70. Page 5 line 223 reference added. (line 236 in last version of manusript)
  71. Page 5 line 225 reference added. (line 238 in last version of manusript)
  72. Page 5 line 228 reference added. (line 241 in last version of manusript)
  73. Page 5 line 230 reference added. (line 243 in last version of manusript)
  74. Page 5 line 231 reference added. (line 244 in last version of manusript)
  75. Page 5 line 236 reference added. (line 249 in last version of manusript)
  76. Page 5 line 240 reference added. (line 254 in last version of manusript)
  77. Page 5 line 243 reference added. (line 256 in last version of manusript)
  78. Page 5 line 252 reference added. (line 265 in last version of manusript)
  79. Page 5 line 254 reference added. (line 267 in last version of manusript)
  80. Page 5 line 256 reference added. (line 270 in last version of manusript)
  81. Page 6 line 269 reference added. (line 283 in last version of manusript)

  1. In line 273, we mentioned that vesicular systems can encapsulate hydrophilic and hydrophobic substances thanks to their chemical structure (aqueous core and lipid bilayer) and lipid-embedded drugs can easily pass through membranes. The authors thank the reviewer for his/her contribution but polymer system reference irrelevant our subject in this line. The vesicular systems discussed in the article are not polymeric but lipid-based systems. The conjuctions link the requested citations at the end of the relevant sentence.
  2. Page 7 line 298 reference added. (line 307 in last version of manusript)
  3. Page 7 line 301 reference added. (line 310 in last version of manusript)
  4. Page 7 line 313 reference added. (line 352 in last version of manusript)
  5. Page 7 line 314 references added. (line 353 in last version of manusript)
  6. Page 7 line 319 reference added. (line 358 in last version of manusript)
  7. Page 7 line 320 reference added. (line 360 in last version of manusript)
  8. Page 7 line 321 reference added. (line 361 in last version of manusript)
  9. Page 8 line 327 reference added. (line 366 in last version of manusript)
  10. Page 8 line 340 2 of CO2 is subscript.
  11. Page 12 line 439 reference added. (line 496 in last version of manuscript)
  12. Page 12 line 440 reference added. (line 498 in last version of manuscript)
  13. Page 12 line 441 reference added. (line 499 in last version of manuscript)
  14. Page 12 line 443 reference added. (line 501 in last version of manuscript)
  15. Page 12 line 446 reference added. (line 505 in last version of manuscript)
  16. Page 12 line 440 2 of CO2 is subscript.
  17. English grammar improved by the reviewers.

Round 3

Reviewer 3 Report

Comments and Suggestions for Authors

The authors improvedtheir manuscript and answered the comments. I suggest the editor to accept this manuscript.